# Epigenetic regulation of mammalian Hedgehog signaling to the stroma determines the molecular subtype of bladder cancer

SungEun Kim[1], Yubin Kim[1], JungHo Kong[1], Eunjee Kim[1], Jae Hyeok Choi[1], Hyeong Dong Yuk[2], HyeSun Lee[2], Hwa-Ryeon Kim[3], Kyoung-Hwa Lee[2], Minyong Kang[4], Jae-Seok Roe[3], Kyung Chul Moon[5], Sanguk Kim[1]*, Ja Hyeon Ku[2]*, Kunyoo Shin[1]*

[1]Department of Life Sciences, Pohang University of Science and Technology, Pohang, Republic of Korea; [2]Department of Urology, Seoul National University Hospital, Seoul, Republic of Korea; [3]Department of Biochemistry, College of Life Science and Biotechnology, Yonsei University, Seoul, Republic of Korea; [4]Department of Urology, Samsung Medical Center, School of Medicine, Sungkyunkwan University, Seoul, Republic of Korea; [5]Department of Pathology, Seoul National University Hospital, Seoul, Republic of Korea

**Abstract** In bladder, loss of mammalian *Sonic Hedgehog* (*Shh*) accompanies progression to invasive urothelial carcinoma, but the molecular mechanisms underlying this cancer-initiating event are poorly defined. Here, we show that loss of *Shh* results from hypermethylation of the CpG shore of the *Shh* gene, and that inhibition of DNA methylation increases *Shh* expression to halt the initiation of murine urothelial carcinoma at the early stage of progression. In full-fledged tumors, pharmacologic augmentation of Hedgehog (Hh) pathway activity impedes tumor growth, and this cancer-restraining effect of Hh signaling is mediated by the stromal response to Shh signals, which stimulates subtype conversion of basal to luminal-like urothelial carcinoma. Our findings thus provide a basis to develop subtype-specific strategies for the management of human bladder cancer.

DOI: https://doi.org/10.7554/eLife.43024.001

*For correspondence:
sukim@postech.ac.kr (SK);
kuuro70@snu.ac.kr (JHK);
kunyoos@postech.ac.kr (KS)

Competing interests: The authors declare that no competing interests exist.

## Introduction

Hedgehog (Hh) signaling has been recognized for its post-embryonic roles in the homeostatic maintenance of tissue integrity and the development of human malignancies (*Ahn and Joyner, 2005*; *Goodrich et al., 1997*; *Shin et al., 2011*; *Taipale and Beachy, 2001*). The initial identification of Hh pathway activity in human cancers, including basal cell carcinoma and medulloblastoma, has led to the development of the first FDA-approved drug targeting the Hh pathway for the treatment of human malignancy (*Goodrich et al., 1997*; *Ruch and Kim, 2013*; *Sekulic et al., 2012*; *Tang et al., 2012*), giving rise to a new field of pharmaceutical intervention (*Teglund and Toftgård, 2010*). Despite promising early preclinical studies (*Olive et al., 2009*; *Yauch et al., 2008*), recent studies investigating pancreatic, colon or ovarian cancers have shown that Hh pathway antagonism is not beneficial and clinical trials had to be halted in some cases because of accelerated cancer growth (*Herter-Sprie et al., 2013*; *Kaye et al., 2012*; *Ruch and Kim, 2013*). Consistent with the results of human trials, several recent studies have shown a protective role of Hh pathway activity in the progression of cancers that originate from endodermally derived tissues, including the bladder

**eLife digest** In order to grow, cancer cells shut down or over-activate genes that normally maintain a cell's health. The *Sonic Hedgehog* gene – named after a Japanese cartoon character – is associated with the cancer of several tissues, including the bladder. In 2014, researchers found that losing the *Sonic Hedgehog* gene, *Shh* for short, is necessary for bladder cancers to become aggressive: *Shh* signals prompt healthy cells near the tumor to inhibit the cancer cell growth, whilst aggressive bladder cancer cells turn off the *Shh* gene. Kim et al. – including many of the researchers involved in the 2014 work – now investigate how cancer cells switch off the *Shh* gene and what effect it has on bladder cancer cells and their surrounding tissue when turned back on.

DNA sequencing bladder cancer cells derived from human patients showed that there were no genetic deletions or mutations within the gene. However, the sequence and nearby regions of DNA did contain methylations – a chemical modification that generally switches genes off. When mice with early stages of bladder cancer were treated with a drug that inhibits methylation, the *Shh* gene turned back on, the bladder cancers stopped growing and the tumors stayed at an early stage of development. When the same drug was used on mice with aggressive bladder cancer, this caused non-cancer cells in the surrounding tissue to respond to Shh and send restraining signals back to the tumor. These signals eventually stopped cancer growth and converted the tumor into a less aggressive type of bladder cancer. Additionally, Kim et al. saw that blocking methylation had the same effect on human bladder cancer cells that had been transplanted into mice.

These results therefore indicate that *Shh* could be a new target for cancer treatments. For instance, drugs that decrease methylation and turn on the gene could be a way of managing cancer in patients with aggressive bladder cancers, which often show low activity of the gene. However, future studies are needed to understand what exactly happens within cancer cells during tumor conversion and to determine if this kind of intervention could have unintended consequences.

DOI: https://doi.org/10.7554/eLife.43024.002

(*Shin et al., 2014a*; *Shin et al., 2014b*), pancreas (*Lee et al., 2014*; *Rhim et al., 2014*), colon (*Gerling et al., 2016*; *Lee et al., 2016*), and prostate (*Yang et al., 2017*). This tumor-restraining effect on a wide range of solid cancers is suggested to be exerted by the stromal response to Hh signals elicited from epithelial cancer cells.

In the bladder, an organ of endodermal origin, the Hh response is restricted to the stroma while Sonic hedgehog (Shh) protein is produced in basal epithelial cells. The regulatory circuit involving Hh signaling feedback between the bladder epithelium and supporting stromal cells is required for the proliferative response to injury during urothelial regeneration (*Shin et al., 2011*). Surprisingly, *Shh* expression is lost during the development of invasive urothelial carcinoma, even though the tumor is derived from Shh-expressing stem cells (*Shin et al., 2014a*), and genetic ablation of Hh signal response in stromal cells accelerates tumor progression at an early stage (*Shin et al., 2014b*). These studies have shown that loss of *Shh* expression invariably accompanies progression to invasive carcinoma and that suppression of the Hh response in the tumor stroma significantly accelerates the initiation of cancer, suggesting that Hh pathway activity protects against tumor progression at early stages of tumor development. Although the expression of *Shh* is invariably lost in both murine and human urothelial carcinoma, deep insights into the mechanisms underlying the tumor-cell-specific regulation of *Shh* at the early stage of carcinogenesis remain elusive. Interestingly, recent large-scale genomic studies in human bladder cancer (*Cancer Genome Atlas Research Network, 2014a*) have revealed that mutations of genes involved in epigenetic regulation are highly enriched in invasive urothelial carcinomas, while an extensive analysis by our group showed no mutational changes in the *SHH* gene; these results raise the possibility that epigenetic activities may be responsible for the loss of *SHH* expression during the initiation of urothelial carcinomas.

Technical advances in cancer genomics have permitted the subdivision of tumors into different molecular subtypes based on gene expression and mutational profiles (*Cancer Genome Atlas Network and Cancer Genome Atlas, 2012*; *Cancer Genome Atlas Research Network, 2015*). Recent large-scale genomic studies of gene expression in human urothelial carcinoma have revealed five distinct subtypes of bladder cancer (*Cancer Genome Atlas Research Network, 2014a*; *Choi et al.,*

*2014*; *Robertson et al., 2017*), which can provide a strategic basis for developing personalized therapeutic interventions for individual patients with different molecular subtypes of urothelial carcinoma with genetic variability. An understanding of cellular and molecular dynamics, however, as cells evolve from the pre-cancerous state to distinct molecular subtypes of invasive carcinoma during tumor progression is required to develop more rationalized and precise treatment options for this malignancy and will require extensive experimental testing and validation through the integrative analysis of subtype-specific tumor initiation and progression beyond phenotypic analysis.

Interactions between epithelial cancer cells and the tumor stroma are important for the initiation and growth of human cancers (*Calon et al., 2015*; *Isella et al., 2015*; *Mao et al., 2013*). Our previous work has shown that Hh signaling to the stroma induces the expression of Bone morphogenetic proteins (BMPs), which impede bladder cancer progression. This anti-cancer effect of Hh-induced stromal expression of BMPs is mediated by urothelial differentiation of pre-cancerous cells at the early stage of tumor initiation (*Shin et al., 2014b*), indicating the importance of the stromal response elicited by tumor cells during tumor progression. Although the protective role of Hh response in the stroma at early stages of tumor initiation is interesting, it remains unknown whether the modulation of the Hh signaling to the stroma, especially increased activity of the Hh pathway, in full-fledged tumors would have a similar antitumor growth effect. This is of particular interest because most patients seen in a clinical setting are at a late stage of disease, with full-grown tumors. In this study, we elucidated the molecular basis for the loss of *Shh* during the development of bladder cancer and showed the role of the Hh signaling response in the tumor stroma in the determination of distinct molecular subtypes and the growth of full-fledged urothelial carcinoma.

## Results

### Loss of *Shh* expression in urothelial carcinoma results from hypermethylation of the CpG shore of the *Shh* gene

Having previously established the absence of *SHH* expression (*Shin et al., 2014b*) with a low incidence of genetic alterations (*Figure 1—figure supplement 1A*) and enrichment of mutations in genes involved in epigenetic regulation in human invasive urothelial carcinomas (*Cancer Genome Atlas Research Network, 2014a*), we compared the level of methylation in the regulatory region of *Shh* between wild-type bladders and N-butyl-N-4-hydroxybutyl nitrosamine (BBN)-induced urothelial carcinomas. By performing bisulfite sequencing analysis, we found significant increases in DNA methylation at the CpG shore upstream of the CpG island of the *Shh* promoter region (*Figure 1—figure supplement 1B*) in murine invasive urothelial carcinomas compared to that in wild-type bladders (*Figure 1A,B*). Pharmacological inhibition of DNA methyltransferase (DNMT) activity with 5'-azacitidine in BBN-induced urothelial carcinoma decreased the level of DNA methylation (*Figure 1A,B*), with significant increases in the expression of *Shh* (*Figure 1C*).

In addition to primary murine tumors, we established 3D bladder tumor organoids derived from BBN-induced urothelial carcinoma (*Figure 1—figure supplement 1C*). Orthotopic transplantation of these tumor organoids revealed the histopathology of parental tumors (*Figure 1—figure supplement 1D*), suggesting that these organoids could recapitulate the pathology of the original BBN-induced urothelial carcinomas. To confirm that the loss of *Shh* expression was due to the increased methylation of the *Shh* gene, murine bladder organoids were cultured and treated with 5'-azacitidine, and the methylation status of the *Shh* regulatory region was analyzed. Consistent with the results from BBN-induced tumors, our analyses revealed that the CpG shore of the *Shh* promoter region in bladder tumor organoids were also hypermethylated, and following the treatment of 5'-azacitidine, the level of methylation was decreased (*Figure 1D,E*), with significant increases in the expression of *Shh* (*Figure 1F*). Our findings in BBN-induced urothelial carcinomas and bladder tumor organoids strongly suggested that the loss of *Shh* expression in invasive urothelial carcinomas results from the hypermethylation of the CpG shore in the *Shh* promoter region.

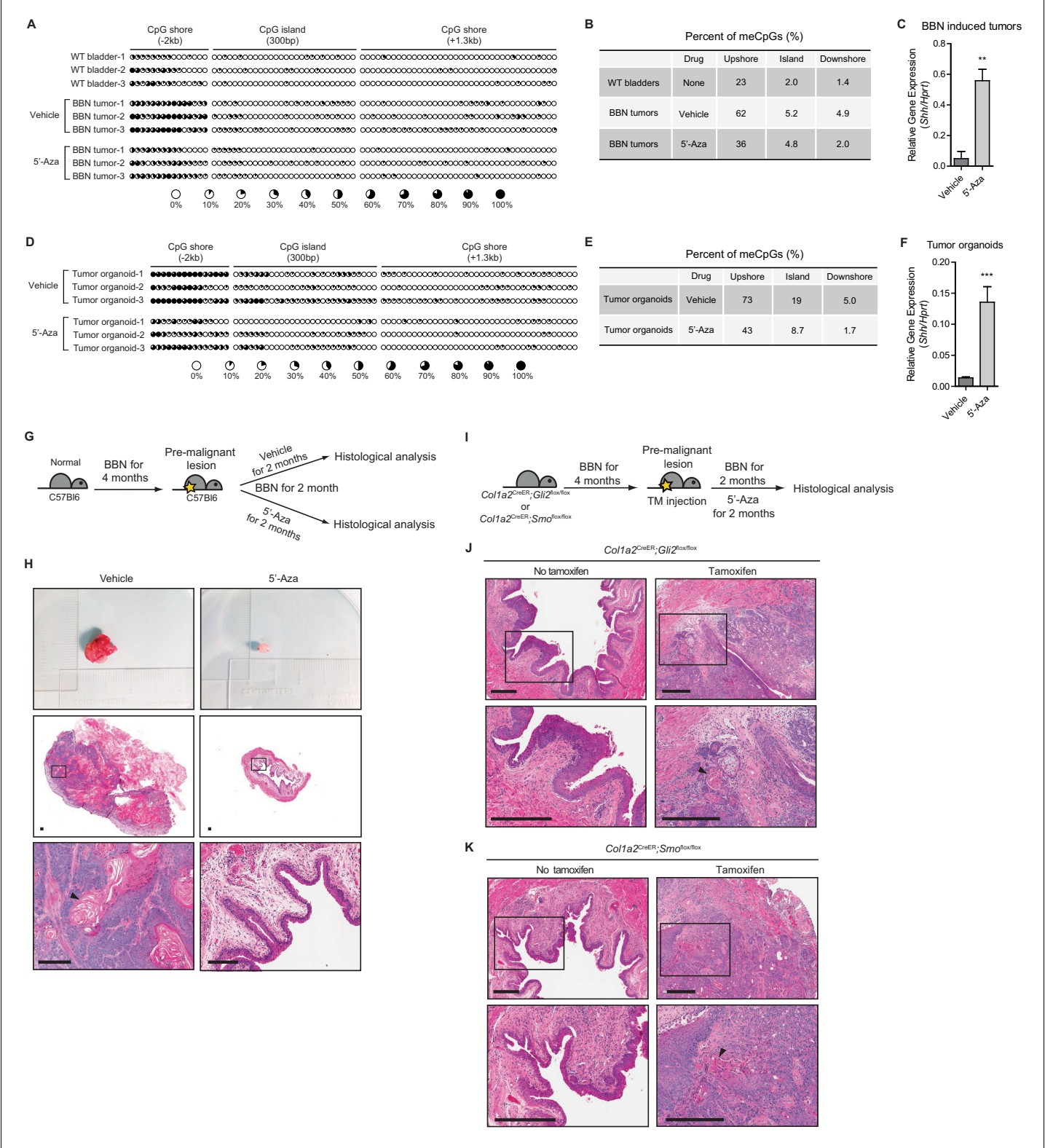

**Figure 1.** Loss of *Shh* expression in urothelial carcinoma due to hypermethylation of the CpG shore of the *Shh* gene. (A, D) The methylation status of the CpG island and CpG shore regions of the *Shh* gene were analyzed by bisulfite sequencing in wild-type bladder tissues, BBN-induced bladder tumors with or without 5'-azacitidine treatment (A), and tumor organoids with or without 5'-azacitidine treatment (D). BBN-induced mouse tumors were orthotopically transplanted and, 1 week after transplantation, the resulting animals were treated with 5'-azacitidine (1 mg per kg mouse body weight) every other day for 2 weeks before methylation analysis. Tumor organoids were cultured using Matrigel overlay method. Three days after seeding,

*Figure 1 continued*

tumor organoids were treated with 5'-azacitidine (1 uM) for 4 consecutive days. Each circle represents one of 81 CpG sites, and the average degree of methylation is indicated by the black portion of the white circle. (B, E) Results obtained from bisulfite sequencing analysis of BBN-induced bladder tumors (B) and tumor organoids (E) are summarized. (C, F) Expression of *Shh* in orthotopically transplanted BBN-induced tumors treated with 5'-azacitidine (C, 11-fold increase) and in cultured tumor organoids treated with 5'-azacitidine (F, 9-fold increase) compared to that of untreated controls. Data are presented as the mean ± SEM, and significance was calculated with an unpaired Student's t test (**, p<0.001). n = 3 technical replicates, and the entire experiment was repeated three times. (G) Schematic diagrams of experimental strategies for evaluating the effect of DNMT inhibition on the initiation of bladder cancer. Mice (14 animals in total) exposed to BBN for 4 months to induce CIS lesions were treated with the vehicle control (seven animals) or 5'-azacitidine (seven animals) for 2 months, with continued BBN exposure to induce the development of invasive carcinoma before histopathological analysis of the bladders. (H) Bladder tumors are shown in the upper panel. H and E staining of bladder sections (middle panels) from mice treated with the vehicle control (left panels) or 5'-azaciditine (right panels). Magnified views (lower panels) of the boxed regions in the middle panels, confirming the presence (vehicle) or absence (5'-azacitidine) of invasive carcinoma. Scale bars represent 150 µm. (I) Schematic diagrams of experimental strategies for testing the association of the stromal Hh response with the anticancer effect of hypomethylation on the initiation of bladder cancer. *Col1a2*CreER;*Gli2*flox/flox (10 animals in total) or *Col1a2*CreER;*Smo*flox/flox (10 animals in total) mice exposed to BBN for 4 months were injected with TM (five animals on each strain) or corn oil (five animals on each strain) on 3 consecutive days. The resulting animals were subsequently exposed to BBN for two additional months with 5'-azacitidine treatment. (J, K) Sections from the bladders of vehicle-injected (left panels) or TM-injected (right panels) mice were analyzed by H and E staining (J, *Col1a2*CreER;*Gli2*flox/flox; K, *Col1a2*CreER;*Smo*flox/flox). Arrowheads in high-magnification images indicate regions of squamous differentiation. Scale bars represent 300 µm. See also *Figure 1—figure supplement 1* and *Figure 1—source data 1*.
DOI: https://doi.org/10.7554/eLife.43024.003

The following source data and figure supplement are available for figure 1:

**Source data 1.** Expression of Shh in BBN-induced tumors and in cultured tumor organoids treated with 5'-azacitidine.
DOI: https://doi.org/10.7554/eLife.43024.005

**Figure supplement 1.** Loss of *Shh* expression in urothelial carcinoma resulting from hypermethylation of the CpG shore of the *Shh* gene increases the initiation of invasive urothelial carcinoma at the premalignant stage of progression through the stromal Hh response.
DOI: https://doi.org/10.7554/eLife.43024.004

## Pharmacological inhibition of DNA methylation halts the initiation of invasive urothelial carcinoma at the premalignant stage of progression through increased Hh signaling to the stroma

Our previous study showed that loss of the stromal Hh response triggers the initiation of invasive urothelial carcinoma and that elevated Hh signaling inhibits the development of bladder cancer at the early stage of progression (*Shin et al., 2014a*; *Shin et al., 2014b*). Given the experimental evidence that the expression of *Shh* is restored under 5'-azacitidine treatment (*Figure 1C,F*), we reasoned that inhibition of DNA methylation activity would impede the development of bladder cancer, especially at the early stage of tumor initiation. To test this possibility, we pharmacologically inhibited DNA methylation with 5'-azacitidine in our BBN-induced bladder cancer model. As previously established (*Shin et al., 2014a*), carcinoma in situ (CIS) became robust and widespread when mice were exposed to BBN for 4 months. This widespread CIS contains premalignant cells with Shh expression and represents a precursor lesion in muscle-invasive carcinoma. To investigate the effect of DNA methylation on bladder cancer initiation, mice were treated with BBN for 4 months, followed by the initiation of treatment with 5'-azacitidine at low dose for an additional 2 months (*Tsai et al., 2012*), while continuing exposure to BBN (*Figure 1G*). Without 5'-azacitidine treatment, BBN exposure for a total of 6 months resulted in the development of invasive carcinoma (*Shin et al., 2014a*; *Shin et al., 2014b*), whereas no invasive carcinoma was observed in mice treated with 5'-azacitidine during the final 2 months (*Figure 1H*, and *Figure 1—figure supplement 1E,F*), suggesting that inhibition of DNA methylation impeded tumor initiation if treatment was administered prior to the formation of invasive carcinoma.

To test whether the anticancer initiation effect of 5'-azacitidine was mediated by an increased stromal Hh response induced by increased expression of Shh in cancer cells, we attempted to rescue the tumor-restraining effect of 5'-azacitidine by genetically suppressing the Hh response in the stroma. As previously reported, Shh expression occurs in basal stem cells in the urothelium and responses to this signal are restricted to stromal cells (*Shin et al., 2011*). To genetically inactivate the stromal Hh response, we used *Col1a2*CreER;*Smo*flox/flox or *Col1a2*CreER;*Gli2*flox/flox mice expressing tamoxifen (TM)-inducible, stroma-specific CreER (*Col1a2*CreER) and carrying homozygous floxed alleles of essential Hh pathway components (Gli2 or Smoothened). These mice were exposed to BBN for 4 months and then injected with TM to genetically ablate Hh response in the stroma prior

to the formation of invasive carcinoma. The mice were then continuously exposed to BBN for an additional two months in the presence of 5'-azacitidine (*Figure 1I*). In mice treated with TM to ablate the Hh response in the stroma, we found that the antitumor initiation effect of 5'-azacitidine was reversed and that invasive carcinomas appeared at 6 months, as in normal BBN-exposed mice, whereas no invasive carcinoma was observed in control animals (*Figure 1J,K* and *Figure 1—figure supplement 1G,H*). These findings suggested that DNA methylation of the *Shh* gene functions as a molecular basis for the loss of *Shh* expression in invasive urothelial carcinoma and confirmed the role of Hh signaling to the stroma in the initiation of bladder cancer at the early stage of disease, as previously reported (*Shin et al., 2014a*; *Shin et al., 2014b*).

## Pharmacological inhibition of DNMT activity impedes the growth of urothelial carcinoma via increased activity of the BMP pathway, induced by the stromal Hh response

Although the increased stromal Hh response induced by inhibiting DNA methylation was shown to inhibit the transition of premalignant lesions to invasive carcinoma at the early stage of tumorigenesis, it remained unclear whether it exerted similar anticancer effects on the growth of mature urothelial carcinomas. To investigate this possibility, we used a recently established orthotopic transplantation model in which bladder cancer cells are intramurally injected into the wall of the bladder dome, allowing the transplanted tumor cells to propagate in physiologically relevant *in vivo* microenvironments (*Shin et al., 2014a*). Mice were orthotopically injected with BBN-induced tumor cells derived from isogenic mice and treated with 5'-azacitidine for 1.5 months after transplantation (*Figure 2A*). In the control group without inhibition of DNA methylation, tumor cells propagated and grew into full-fledged invasive carcinomas (*Figure 2B* and *Figure 2—figure supplement 1A*). In bladders from 5'-azacitidine-treated mice, however, no invasive carcinoma was observed (*Figure 2B* and *Figure 2—figure supplement 1B*), suggesting that inhibition of DNA methylation fully impeded the growth of bladder tumors in immunocompetent wild-type animals.

To investigate whether the anticancer propagation effect of 5'-azacitidine was mediated by the stromal Hh response, we combined our pharmacological approach using 5'-azacitidine with a genetic approach to genetically suppress the stromal Hh response while pharmacologically increasing the expression of *Shh* in tumor cells (*Figure 2C*). To genetically inactivate the stromal Hh response in recipient mice, we used the *Col1a2*$^{CreER}$;*Gli2*$^{flox/flox}$ and *Col1a2*$^{CreER}$;*Smo*$^{flox/flox}$ strains. After 5 days of recovery from TM injection, BBN-induced tumors, which are derived from mice with an isogenic background, were orthotopically transplanted, followed by treatment with 5'-azacitidine for 1.5 months (*Figure 2C*). In both strains with genetic ablation of the stromal Hh response, the anticancer growth effect of 5'-azacitidine disappeared (*Figure 2D,E*). These results strongly suggested that the effect of 5'-azacitidine on the propagation of tumor cells was mediated through the stromal Hh response elicited by Shh, whose expression is regulated epigenetically by cancer cells.

Next, we sought to determine whether the Hh signaling-mediated, antitumor propagation effect was regulated by Bmp, a secreted stromal factor whose expression is known to be regulated by the stromal Hh response in bladder (*Shin et al., 2014b*). Bmps are secreted stromal factors for urothelial differentiation (*Mysorekar et al., 2009*), and BMP pathway activity impedes bladder cancer progression prior to the formation of invasive carcinoma by stimulating urothelial differentiation (*Shin et al., 2014b*). However, the role of stromal Bmp in later stages of tumor development, especially in the tumor growth, is unknown. To determine whether stromal Hh response-regulated Bmp expression is involved in bladder cancer growth, we overexpressed Bmp4 in bladder tumor organoids derived from BBN-induced tumors (*Figure 1—figure supplement 1C,D*). The expression of Bmp4 in these organoids was increased by 10-fold compared with that in control organoids (*Figure 2—figure supplement 1C*). The resulting organoids with Bmp4 expression were orthotopically injected into *Col1a2*$^{CreER}$;*Smo*$^{flox/flox}$ and *Col1a2*$^{CreER}$;*Gli2*$^{flox/flox}$ mice, and the mice were then injected with TM to genetically ablate the stromal Hh response. These animals were subsequently treated with 5'-azacitidine for 1 month to increase *Shh* expression in tumor cells (*Figure 3A*). We found that, compared with wild-type bladder tumor organoids, Bmp4-expressing tumor organoids showed growth reductions after transplantation (*Figure 3B,C* and *Figure 2—figure supplement 1D,E*). In addition, after tumor organoids were cultured in the presence of the Bmp4 protein to increase the BMP response (*Figure 2—figure supplement 1F*), the Bmp4-treated organoids grew slowly, and the efficiency of organoid formation was significantly reduced (*Figure 3D,E,F*). Taken together, the tumor-

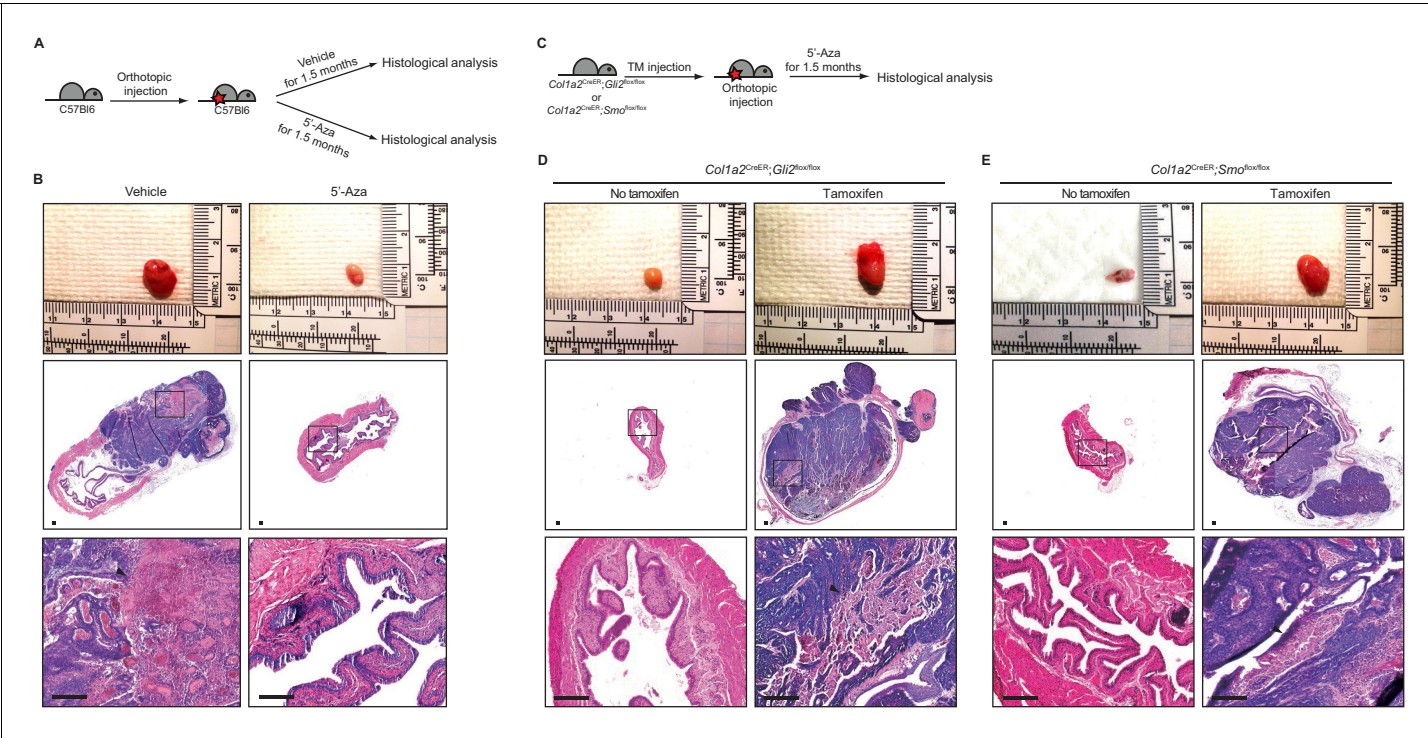

**Figure 2.** Pharmacological inhibition of DNMT activity impedes the growth of urothelial carcinoma through an increased stromal Hh response. (A) Schematic diagrams of the experimental strategies for evaluating the effect of DNMT inhibition on the growth of bladder cancer. Mice (14 animals in total) orthotopically injected with BBN-induced bladder tumor cells were treated with the vehicle control (seven animals) or 5'-azacitidine (seven animals) for 1.5 months. (B) Orthotopic allografts of BBN-induced tumors are shown in the upper panels. H and E staining of allograft sections from mice treated with the vehicle control or 5'-azacitidine is shown in the middle panels. Magnified views (lower panels) of the boxed regions, confirming the presence (vehicle) or absence (5'-azacitidine) of invasive carcinoma. (C) Schematic diagrams of the experimental strategies for testing the association of the stromal Hh response with the anticancer effect of hypomethylation on the growth of bladder cancer. $Col1a2^{CreER};Gli2^{flox/flox}$ or $Col1a2^{CreER};Smo^{flox/flox}$ mice were injected with TM on three consecutive days. BBN-induced tumors from isogenic mice were orthotopically transplanted, and treatment with 5'-azacitidine was initiated for 1.5 months (D, E) Orthotopic allografts of BBN induced tumors are shown in the upper panels. Sections of allografts from vehicle-injected (left panels) or TM-injected (right panels) mice were analyzed by H and E staining (D, $Col1a2^{CreER};Gli2^{flox/flox}$; E, $Col1a2^{CreER};Smo^{flox/flox}$). H and E staining of tumor sections is shown in the middle panels. The lower panels show magnified views of the boxed regions. Arrowheads in high-magnification images indicate regions of squamous differentiation. Scale bars represent 150 μm. See also *Figure 2—figure supplement 1A,B and C*.

DOI: https://doi.org/10.7554/eLife.43024.006

The following figure supplement is available for figure 2:

**Figure supplement 1.** Pharmacological inhibition of DNMT activity impedes the growth of urothelial carcinoma by increasing Hh/BMP signaling activity.

DOI: https://doi.org/10.7554/eLife.43024.007

suppressing effect of Bmp, particularly *in vivo* where tumor cells still express Shh (with 5'-azacitidine treatment) while the stromal Hh response is suppressed (using $Col1a2^{CreER};Smo^{flox/flox}$ and $Col1a2^{CreER};Gli2^{flox/flox}$ mice), supports a potential scenario of an increased reciprocal tumor-stromal signal feedback loop in which hypomethylation-induced Shh secretion by tumor cells activates the Hh response in bladder stroma, resulting in stromal expression of Bmps, which in turn signal back to tumor cells to impede their growth.

## Heightened activity of Hh signaling to the stroma induces a less aggressive luminal subtype of urothelial carcinoma

To investigate the cellular basis of the cancer-restraining effects of the stromal Hh response induced by Shh, which is regulated by DNA methylation in tumor cells, on the growth of bladder tumors, we performed the orthotopic transplantation of BBN induced tumors to nude mice. These mice were chosen to grow transplanted tumors under more permissive conditions because orthotopic

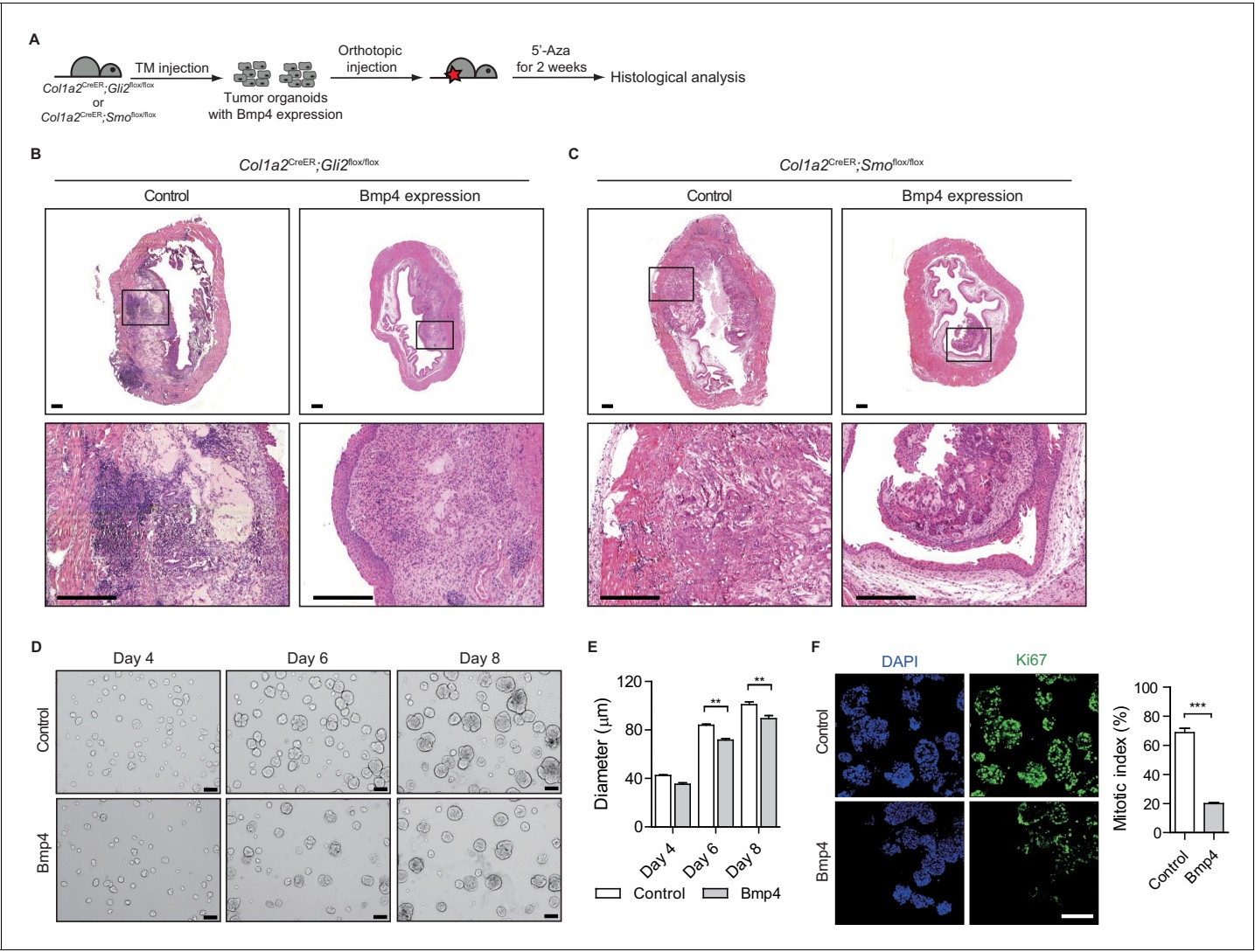

**Figure 3.** Pharmacological inhibition of DNMT activity impedes the growth of urothelial carcinoma by increasing BMP pathway activity. (**A**) Experimental scheme for evaluating the role of BMP signaling in the growth of urothelial carcinoma. TM was injected into *Col1a2*[CreER]*;Gli2*[flox/flox] (eight animals in total) or *Col1a2*[CreER]*;Smo*[flox/flox] (eight animals in total) mice on 3 consecutive days. Mice were then orthotopically injected with Bmp4-expressing bladder tumor organoids and subsequently treated with 5'-azacitidine for 2 weeks. (**B, C**) Sections of allografts from mice orthotopically injected with control tumor organoids (left panels) or Bmp4-expressing tumor organoids (right panels) were stained with H and E. The lower panels represent magnified views of the boxed region in the middle panels. Scale bars represent 300 µm. (**D**) Tumor organoids derived from BBN-induced bladder tumors were cultured in the absence (upper panels) or presence (lower panels) of Bmp4 for 8 days. Left, middle and right panels show the bright-field images of tumor organoids cultured for 4, 6 and 8 days, respectively. Scale bars represent 100 µm. (**E**) Average size of bladder tumor organoids cultured for 4, 6, and 8 days in the absence or presence of Bmp4 protein (n = 90 organoids in each condition). (**F**) Quantification of cell proliferation in tumor organoids cultured for 6 days in the absence or presence of Bmp4. Left panels show the images immunostained with DAPI and Ki67. Ki67-positive cells are shown as a per cent of total DAPI-staining nuclei. Data are presented as the mean ± SEM, and significance was calculated with an unpaired Student's t test (**, p<0.01). See also *Figure 2—figure supplement 1D,E,F* and *Figure 3—source data 1*.
DOI: https://doi.org/10.7554/eLife.43024.008
The following source data is available for figure 3:

**Source data 1.** Quantification for the growth and cell proliferation of tumor organoids with Bmp4 treatment.
DOI: https://doi.org/10.7554/eLife.43024.009

transplantation of bladder tumors to wild-type mice in the presence of 5'-azacitidine led to complete blockade of tumor growth (*Figure 2A,B*); thus, the use of nude mice under more permissive conditions for tumor growth allowed us to overcome the difficulty in studying the basis of the anticancer effect of the hypomethylation-induced stromal Hh response on tumor growth. Indeed, transplanted

bladder tumors in nude mice grew even under 5'-azacitidine treatment, but they gave rise to tumor lesions with smaller sizes than those without 5'-azacitidine treatment (*Figure 4A–C* and *Figure 4—figure supplement 1A,B*). These findings suggest that 5'-azacitidine treatment is still effective in suppressing tumor growth under immunocompromised conditions, which is consistent with our earlier results (*Figure 2A,B*).

Next, we sought to investigate the molecular and cellular basis for the less aggressive growth of bladder tumors when Hh signaling activity is increased by inhibiting DNA methylation. As shown previously (*Figures 2* and *3*), the anticancer effects of Hh signaling appeared to be mediated by stromal BMP, whose signaling activity is known to be associated with the urothelial differentiation of basal cells into luminal cells (*Mysorekar et al., 2009*). As our previous study on the cellular origin of bladder cancer showed that urothelial carcinoma is derived from basal stem cells (*Shin et al., 2014a*), we hypothesized that increased Hh signaling activity might cause the tumor to differentiate into a less aggressive form of luminal subtype, leading to much slower growth upon 5'-azacitidine treatment. As previously reported (*Fantini et al., 2018*; *Shin et al., 2014b*), based on the expression level of basal markers and mutational profile, the invasive carcinomas produced in our BBN model are most similar to the basal subtype of human urothelial carcinoma (*Figure 4—figure supplement 1C*), which is the most aggressive form of bladder cancer (*Choi et al., 2014*; *Vasconcelos-Nóbrega et al., 2012*). Thus, we examined BBN-induced bladder tumors that were orthotopically grown in the presence of 5'-azacitidine to investigate the cellular differentiation of transplanted tumors. Our immunohistochemical analysis revealed that the expression of the luminal marker, Ck18, was increased in the tumors treated with 5'-azacitidine, while control bladders displayed basal phenotypes such as squamous differentiation and the expression of Ck5, a marker for the basal subtype (*Figure 4D,E* and *Figure 4—figure supplement 1D,E*). Moreover, our quantitative RT-PCR experiments showed that the expression level of luminal markers relative to basal markers was significantly increased in the transplanted bladder tumors under 5'-azacitidine treatment compared with that of the control group, which was accompanied by increased expression of *Shh* (*Figure 4F* and *Figure 4—figure supplement 1F*). Consistent with these results, our analysis of RNA-seq expression profiles revealed the strong luminal signature with relatively low expression of basal markers in the tumors orthotopically grown in the presence of 5'-azacitidine, whereas control allografts in the absence of 5'-azacitidine showed clear standard signature of basal subtype (*Figure 4G* and *Figure 4—figure supplement 1G*) (*Damrauer et al., 2014*). These results suggested that subtype conversion between basal and luminal-like subtypes resulting from the increased Hh response induced by epigenetically upregulated expression of *Shh* in tumor cells might account for the reduced tumor growth.

## Stromal Hh response-induced conversion from the basal to luminal-like subtype requires BMP pathway activity

To investigate whether conversion from the basal to luminal-like subtype upon 5'-azacitidine treatment is mediated by elevated expression of Hh in tumor cells, we performed orthotopic transplantation of BBN-induced tumor organoids engineered to express shRNA targeting *Shh* (*Figure 5A* and *Figure 5—figure supplement 1A*). The resulting organoids maintained low levels of *Shh* expression after transplantation, even upon 5'-azacitidine treatment, whereas the control organoids showed increased expression of *Shh* (*Figure 5—figure supplement 1B*). The expression levels of luminal markers were significantly decreased in tumor organoids following the expression of the shRNA targeting *Shh* after transplantation, indicating the basal-like characteristics including squamous differentiation of the resulting tumors, even under 5'-azacitidine treatment (*Figure 5B,C* and *Figure 5—figure supplement 1C,D*). Consistent with these results, our analysis of RNA-seq expression profiles confirmed that the gene signature related with the luminal status is negatively enriched in the tumors expressing shRNA for *Shh*, whereas tumor expressing control shRNA showed standard signature of luminal-like subtype (*Figure 5D* and *Figure 5—figure supplement 1F*). Similar results were noted when tumor organoids expressing shRNA targeting *Shh*, marked by mCherry expression, were mixed and transplanted together with control organoids, marked by EGFP, into the same *in vivo* microenvironment (*Figure 5F*). The resulting allografts revealed that mCherry-labeled tumors developed into a more aggressive, rapidly growing basal-like subtype, whereas, in the same microenvironment, EGFP-labeled tumors developed into a less aggressive luminal-like subtype following treatment with 5'-azacitidine (*Figure 5G* and *Figure 5—figure supplement 2A*), demonstrating Hh-mediated conversion of the bladder tumor subtype.

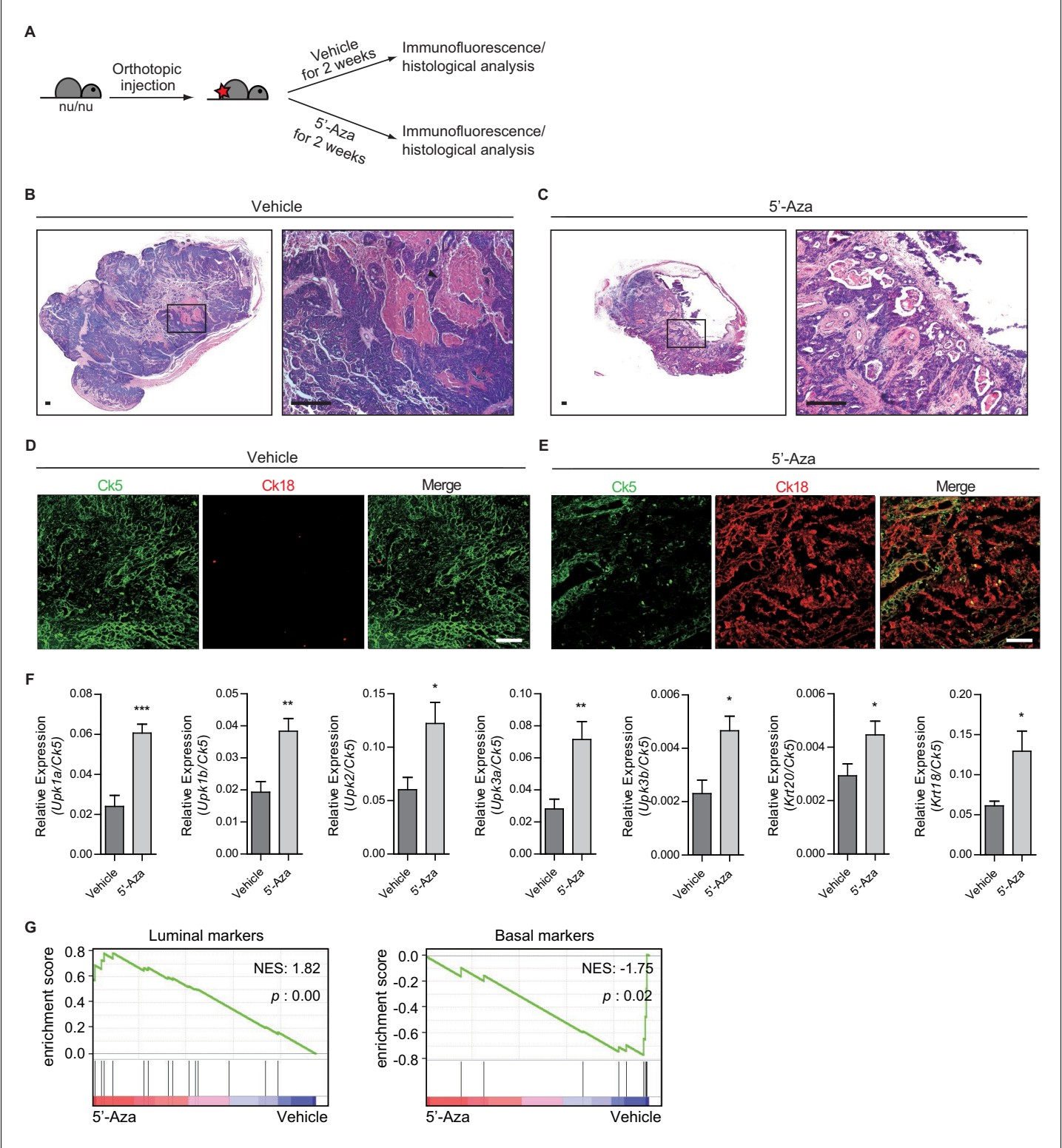

**Figure 4.** Heightened activity of Hh signaling to the stroma induces a less aggressive luminal-like subtype of urothelial carcinoma. (**A**) Schematic diagrams of experimental strategies for evaluating the effect of DNMT inhibition on the growth of bladder cancer under immunocompromised conditions. Nude mice (14 animals in total) orthotopically injected with BBN-induced bladder tumor cells were treated with the vehicle control (seven animals) or 5'-azacitidine (seven animals) for 2 weeks. (**B, C**) H and E staining of allograft sections from mice treated with the vehicle control (**B**) or 5'-azaciditine (**C**) is shown in the left panels. Right panels show magnified views of the boxed regions in the left panels. Arrowhead in the high-magnification image indicates the region of squamous differentiation. Scale bars represent 150 μm. (**D, E**) Sections of tumor allografts from mice

*Figure 4 continued on next page*

*Figure 4 continued*

injected the vehicle control (D) or 5'-azacitidine (E) were immunostained for the basal and luminal markers, Ck5 (green) and Ck18 (red), respectively. Scale bars represent 100 µm. (F) Expression of the luminal markers, *Upk1a* (3-fold increase), *Upk1b* (2-fold increase), *Upk2* (2-fold increase), *Upk3a* (2.5-fold increase), *Upk3b* (2-fold increase), *Krt20* (1.5-fold increase), and *Krt18* (2-fold increase) in tumor allografts from mice with the vehicle control or 5'-azaciditine treatment. Gene expression was normalized to a basal marker (Krt5). Data are presented as the mean ± SEM, and significance was calculated with an unpaired Student's t test (*, p<0.05; **, p<0.01; ***, p<0.001). n = 3 technical replicates, and the entire experiment was repeated six times. (G) Gene set enrichment analysis (GSEA) of tumor allografts treated with the vehicle control (shown in panel B) and 5'-azacitidine (shown in panel C) from RNA-Seq data using standard luminal and basal signatures obtained from previous studies. Normalized enrichment score (NES) and nominal p-value (p) were provided from GSEA accordingly. See also *Figure 4—figure supplement 1* and *Figure 4—source data 1*.

DOI: https://doi.org/10.7554/eLife.43024.010

The following source data and figure supplement are available for figure 4:

**Source data 1.** Relative expression of the luminal markers to a basal marker in tumor allografts from mice with 5'-azaciditine treatment.
DOI: https://doi.org/10.7554/eLife.43024.012
**Figure supplement 1.** Hh signaling to the stroma induces a less aggressive luminal-like subtype of urothelial carcinoma.
DOI: https://doi.org/10.7554/eLife.43024.011

To further determine whether conversion between the basal and luminal-like subtypes requires Hh-mediated BMP signaling, whose activity is necessary for the suppression of tumor growth (*Figure 3*), we established transplant models in which BBN-induced tumor organoids transduced to express shRNA targeting *Bmpr1a* were orthotopically injected into the mouse bladder (*Figure 5A* and *Figure 5—figure supplement 1A*). The expression of *Bmpr1a* was significantly decreased in the resulting tumor organoids (*Figure 5—figure supplement 1B*), and transplantation of *Bmpr1a* knockdown tumor organoids in the presence of 5'-azacitidine gave rise to secondary tumors with decreased expression of luminal markers with squamous differentiation compared to tumor organoids with intact *Bmpr1a* (*Figure 5B,C* and *Figure 5—figure supplement 1C,E*). Consistent with these results, our analysis of RNA-seq expression profiles confirmed that the gene signature related with the luminal status is negatively enriched in the tumors expressing shRNA for *Bmpr1a* (*Figure 5E* and *Figure 5—figure supplement 1G*), whereas tumor expressing control shRNA showed standard signature of luminal-like subtype. Similar data were obtained when tumor organoids expressing shRNA targeting *Bmpr1a*, marked by mCherry expression, were mixed and transplanted together with control organoids, marked by EGFP, into the same *in vivo* microenvironment (*Figure 5F*). The mixed allografts showed that mCherry-labeled tumors developed into a more rapidly growing basal-like subtype than EGFP-labeled tumors with the luminal-like subtype, even under 5'-azacitidine treatment (*Figure 5H* and *Figure 5—figure supplement 2B*). Furthermore, we genetically ablated *Bmpr1a* from BBN-induced tumor organoids derived from *Bmpr1a*<sup>flox/flox</sup> mice (*Mishina et al., 2002*) by expressing Cre recombinase (*Figure 5—figure supplement 3A,B*). Consistent with the findings described above, the resulting organoids developed into basal invasive carcinoma, even with 5'-azacitidine treatment (*Figure 5—figure supplement 3C,D*).

Taken together, our results from various genetic experiments in combination with pharmacological approaches to manipulate Hh and BMP signaling feedback during the growth of bladder tumors strongly suggest that subtype conversion between basal and luminal-like subtypes depends on the reciprocal signaling feedback between tumor cells and the stroma involving epigenetically regulated epithelial Hh expression in tumor cells, stromal Hh response-induced Bmp expression, and the BMP response in tumor cells.

## Increased methylation of the *SHH* gene induces the basal subtype of human urothelial carcinoma and promotes tumor growth through decreased activity of Hh/BMP signaling feedback between cancer cells and tumor stroma

To determine whether Hh/BMP signaling feedback between the tumor and stroma could control the growth of tumors and determines their subtype in human bladder cancer, we first examined multiple bladder cancer cell lines derived from human invasive urothelial carcinomas. Three invasive bladder cancer cell lines, J82, T24 and TCC-SUP, were chosen (*Bubeník et al., 1973*; *Nayak et al., 1977*; *O'Toole et al., 1978*) and analyzed for the level of methylation in the regulatory region of *SHH* by bisulfite sequencing. Consistent with our mouse experiments described above (*Figure 1*), all three

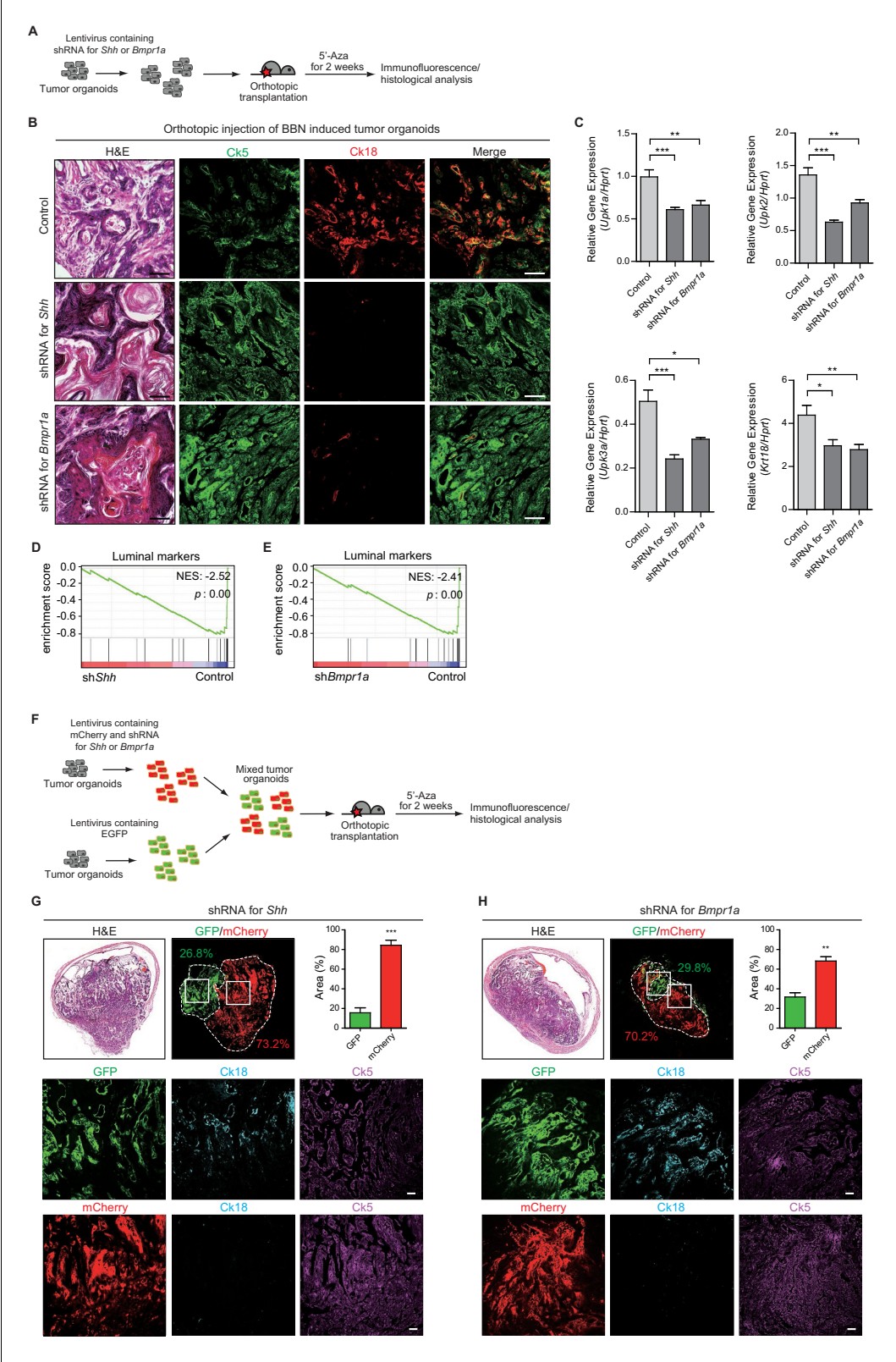

**Figure 5.** Subtype conversion of basal to luminal-like urothelial carcinoma requires Hh and BMP signaling feedback between the tumor and stroma. (**A**) Tumor organoids derived from BBN-induced bladder tumors were infected using a lentivirus containing shRNA targeting *Shh* or *Bmpr1a*. The resulting organoids were injected intramurally into the dome of the bladder, and the mice (15 animals in total) were treated with 5'-azacitidine for 2 weeks. (**B**) Sections of allografts from mice orthotopically injected with control tumor organoids (upper panels, five animals), organoids with shRNA targeting *Shh*

*Figure 5 continued on next page*

*Figure 5 continued*

(middle panels, five animals) or organoids with shRNA targeting *Bmpr1a* (lower panels, five animals) were stained with H and E. Serial sections were immunostained for Ck5 (green) and Ck18 (red). Arrowheads indicate regions of squamous differentiation. Scale bars represent 100 µm. (C) Expression of the luminal markers, *Upk1a* (shRNA for *Shh*, 1.6-fold decrease; shRNA for *Bmpr1a*, 1.5-fold decrease), *Upk2* (shRNA for *Shh*, 2-fold decrease; shRNA for *Bmpr1a*, 1.5-fold decrease), *Upk3a* (shRNA for *Shh*, 2 -fold decrease; shRNA for *Bmpr1a*, 1.6-fold decrease), and *Krt18* (shRNA for *Shh*, 1.5-fold decrease; shRNA for *Bmpr1a*, 1.5-fold decrease) in tumor allografts from mice injected with control tumor organoids or tumor organoids carrying shRNA targeting *Shh* or *Bmpr1a*. Data are presented as the mean ± SEM, and significance was calculated with an unpaired Student's t test (*, p<0.05; **, p<0.01; ***, p<0.001). n = 3 technical replicates, and the entire experiment was repeated five times. (D, E) Gene set enrichment analysis (GSEA) of tumor allografts expressing control shRNA and shRNA targeting *Shh* (D), or shRNA targeting *Bmpr1a* (E) from RNA-Seq data using standard luminal signatures obtained from previous study. Normalized enrichment score (NES) and nominal p-value (p) were provided from GSEA accordingly. (F) Tumor organoids were infected using a lentivirus containing control shRNA with EGFP or shRNA targeting *Shh* or *Bmpr1a* with mCherry. The same number of each resulting organoids were picked manually, mixed and orthotopically transplanted into the nude mice. The mice (eight animals in total) were then treated with 5'-azacitidine for 2 weeks. (G, H) Sections of allografts from mice orthotopically injected with mixed organoids (G, organoid with shRNA targeting *Shh*, four animals; H, organoids with shRNA targeting *Bmpr1a*, four animals) were analyzed by H and E staining or by immunostaining for EGFP, mCherry, Ck18 (cyanine, pseudo) and Ck5 (magenta, pseudo). EGFP or mCherry-positive tumor areas are outlined with dotted lines, and each area was measured and quantified using the Image J program. The middle and lower panels represent magnified views of the boxed regions in the upper panels. Scale bars represent 50 µm. Data are presented as the mean ± SEM, and significance was calculated with an unpaired Student's t test. (**, p<0.01; ***, p<0.001). n = 4 biological replicates. See also *Figure 5—figure supplements 1*, *2* and *3* and *Figure 5—source data 1*.
DOI: https://doi.org/10.7554/eLife.43024.013

The following source data and figure supplements are available for figure 5:

**Source data 1.** Expression of the luminal markers in tumor allografts expressing shRNA targeting Shh, or Bmpr1a, and the quantification of EGFP (expressing control shRNA), or mCherry (expressing shRNA for Shh or Bmpr1a)-positive tumor areas.
DOI: https://doi.org/10.7554/eLife.43024.017
**Figure supplement 1.** Subtype conversion of basal to luminal-like urothelial carcinoma requires Hh and BMP signaling feedback.
DOI: https://doi.org/10.7554/eLife.43024.014
**Figure supplement 2.** Slow tumor growth and subtype conversion require heightened Hh/BMP signaling feedback between the tumor and stroma.
DOI: https://doi.org/10.7554/eLife.43024.015
**Figure supplement 3.** Blockade of BMP signaling impedes subtype conversion of urothelial carcinoma.
DOI: https://doi.org/10.7554/eLife.43024.016

cell lines showed a significant increase in DNA methylation at the CpG shore of the *SHH* promoter region (*Figure 6A,B* and *Figure 6—figure supplement 1A*) and demonstrated basal characteristics of urothelial carcinoma with low levels of *SHH* expression. Pharmacological inhibition of DNMT activity with 5'-azacitidine in these cell lines significantly decreased the level of DNA methylation (*Figure 6A,B*), which was associated with marked increases in the expression of *SHH* (*Figure 6C*). Interestingly, a cell line derived from the luminal-papillary subtype RT4 (*Rigby and Franks, 1970*), exhibiting high expression of *SHH,* showed no significant changes in the expression of *SHH*, even with the DNMT inhibition (data not shown).

To investigate the functional roles of SHH expression in the growth of human bladder tumors and the effects of Hh/BMP signaling feedback between the tumor and stroma on the subtype determination of human invasive urothelial carcinoma, we established an orthotopic xenograft model in which one human invasive bladder cancer cell lines J82 was transplanted into immunocompromised mice (Nod/Scid/Rag2), followed by 5'-azacitidine treatment for 1 month (*Figure 6D*). In the control group without inhibition of DNA methylation, tumor cells developed into full-fledged invasive carcinomas (*Figure 6E* and *Figure 6—figure supplement 1B*). In bladders from 5'-azacitidine treated mice, however, much smaller lesions were observed (*Figure 6E* and *Figure 6—figure supplement 1C*), suggesting that inhibition of DNA methylation inhibited the growth of human bladder tumor. In addition, xenografts from 5'-azacitidine-treated animals showed characteristics of the luminal-like subtype, with increased expression of luminal markers (*Figure 6F*), consistent with our observations in the above-described mouse model (*Figure 4*).

To further confirm the requirement for Hh/BMP signaling feedback in the subtype determination of human bladder cancer, we performed xenograft transplantation of a human bladder cancer cell line that was engineered to express shRNA targeting *SHH* or *BMPR1A* (*Figure 6G*). After transplantation, the resulting tumors maintained a low level of *SHH* or *BMPR1A* expression (*Figure 6H*), and the expression levels of basal markers were significantly increased, with low expression levels of luminal markers being observed even upon 5'-azacitidine treatment (*Figure 6I*). These human data

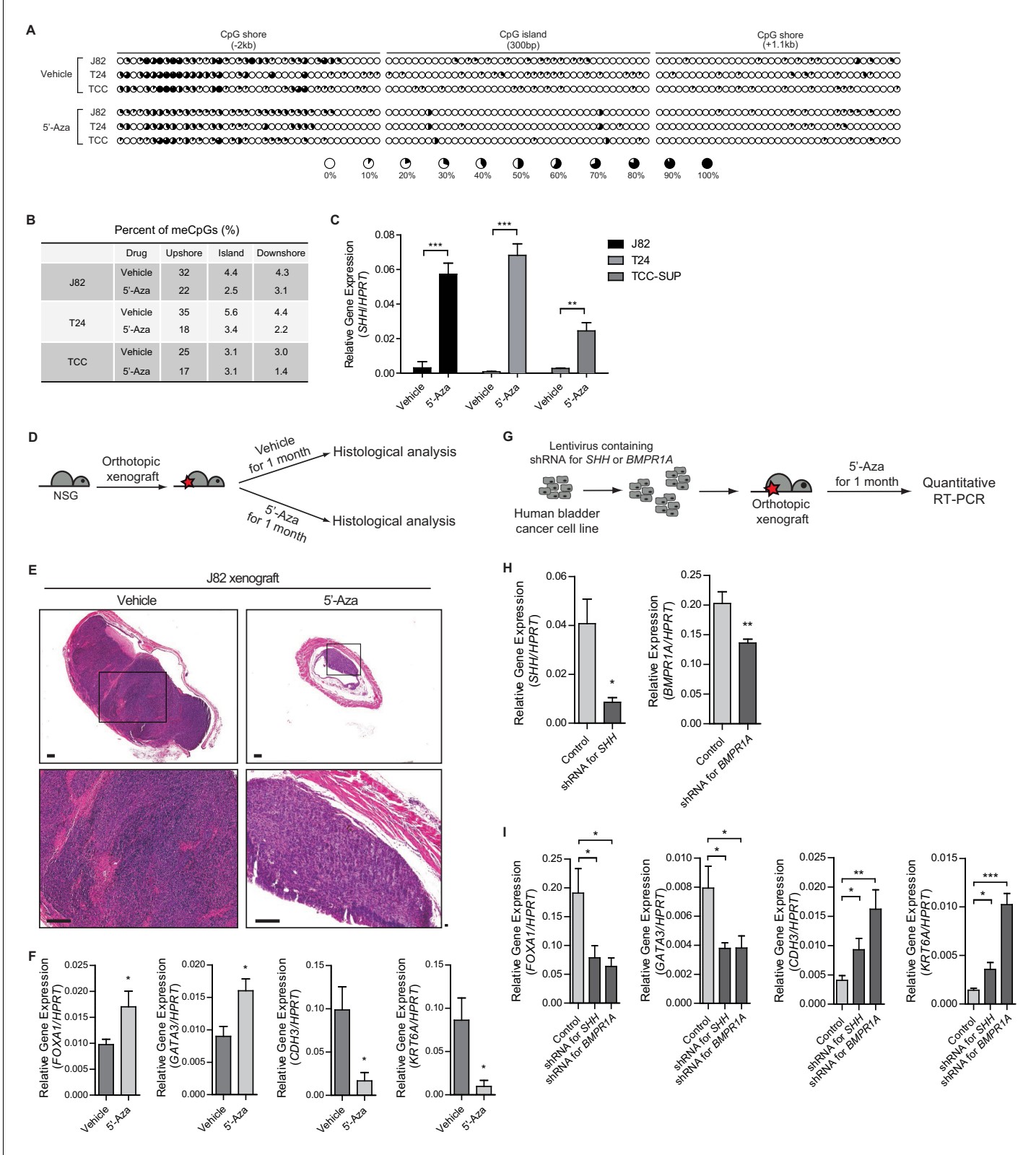

**Figure 6.** Increased methylation of the *SHH* gene induces the basal subtype of human urothelial carcinoma through decreased activity of Hh/BMP signaling feedback between cancer cells and the tumor stroma. (**A**) The methylation status of the CpG island and CpG shore regions of the human *SHH* gene was analyzed by bisulfite sequencing in three human invasive urothelial carcinoma cell lines, J82, T24, and TCC-SUP with or without 5′-azacitidine treatment. Each circle represents one of 117 CpG sites, and the average degree of methylation is indicated by the black portion of the white circle. (**B**)

*Figure 6 continued on next page*

*Figure 6 continued*

The results obtained from bisulfite sequencing analysis of (**A**) are summarized. (**C**) Expression of *SHH* in J82, T24, and TCC-SUP treated with 5'-azacitidine (J82, 6-fold increase; T24, 7-fold increase; TCC-SUP, 3-fold increase) compared to that of untreated controls. Data are presented as the mean ± SEM, and significance was calculated with an unpaired Student's t test (**, $p<0.01$; ***, $p<0.001$). n = 3 technical replicates, and the entire experiment was repeated three times. (**D**) Orthotopic xenograft of J82 cells in immunocompromised (NOD/SCID/IL2Rgnull) mice (14 animals in total), followed by 5'-azacitidine treatment for 1 month. (**E**) H and E staining of orthotopic xenograft sections from mice treated with the vehicle control (left panels, seven animals) or 5'-azacitidine (right panels, seven animals). The lower panels show magnified views of the boxed regions in the upper panels. Scale bars represent 300 µm. (**F**) Expression of the luminal markers, *FOXA1* (1.8-fold increase) and *GATA3* (1.8-fold increase), and the basal markers, *CDH3* (6-fold decrease) and *KRT6A* (9-fold decrease) in tumor xenografts from mice treated with 5'-azacitidine compared to those of the vehicle control. Data are presented as the mean ± SEM, and significance was calculated with an unpaired Student's t test (*, $p<0.05$). n = 3 technical replicates, and the entire experiment was repeated six times. (**G**) The J28 cell line was infected using a lentivirus containing shRNA targeting *SHH* or *BMPR1A*. The resulting cell line was orthotopically xenografted into the dome of the bladder, and the mice were treated with 5'-azacitidine for 1 month. (**H**) Expression of *SHH* or *BMPR1A* in tumor xenografts from mice injected with J82 carrying shRNA targeting *SHH* or *BMPR1A*, respectively. Data are presented as the mean ± SEM, and significance was calculated with an unpaired Student's t test (*, $p<0.05$; **, $p<0.01$). (**I**) Expression of the luminal markers, *FOXA1* (shRNA for *SHH*, 2.5-fold decrease; shRNA for *BMPR1A*, 3-fold decrease) and *GATA3* (shRNA for *SHH*, 2-fold decrease; shRNA for *BMPR1A*, 2-fold decrease), and the basal markers, *CDH3* (shRNA for *SHH*, 2.3-fold increase; shRNA for *BMPR1A*, 4-fold increase) and *KRT6A* (shRNA for *SHH*, 2.5-fold increase; shRNA for *BMPR1A*, 7.2-fold increase) in tumor xenografts from mice injected with control J82 or J82 carrying shRNA targeting *SHH* or *BMPR1A*. n = 3 technical replicates, and the entire experiment was repeated six times. Data are presented as the mean ± SEM, and significance was calculated with an unpaired Student's t test (*, $p<0.05$; **, $p<0.01$; ***, $p<0.001$). See also *Figure 6—figure supplement 1A,B,C* and *Figure 6—source data 1*.

DOI: https://doi.org/10.7554/eLife.43024.018

The following source data and figure supplement are available for figure 6:

**Source data 1.** Expression of *SHH*, luminal and basal markers in bladder tumor cell lines and in tumor xenografts from mice treated with 5'-azacitidine, or expressing shRNA for *SHH* or *BMPR1A*.

DOI: https://doi.org/10.7554/eLife.43024.020

**Figure supplement 1.** Expression analyses of patient samples and large-scale transcriptional data suggest that the decreased methylation of the CpG shore of the human *SHH* promoter region impedes the growth of human urothelial carcinoma by inducing a luminal-like subtype through increased Hh/BMP signaling feedback.

DOI: https://doi.org/10.7554/eLife.43024.019

are consistent with our observations in the mouse model, showing that the Hh pathway response in the tumor stroma induced by elevated expression of Hh in tumor cells impedes the growth of bladder cancer through stromal BMP-induced subtype conversion.

## Expression analysis of patient-derived urothelial carcinomas and large-scale transcriptional analyses suggest an association of the basal subtype of human urothelial carcinoma with poor patient survival, due to the loss of Hh signaling activity

To validate the role of the loss of SHH induced by hypermethylation in the determination of the molecular subtype of human bladder cancer, we examined 10 muscle-invasive urothelial carcinoma samples derived from patients (*Table 1*). We found that the invasive carcinoma samples from six patients showed characteristics of the basal subtype, with the expression of basal markers being significantly increased (*Figure 7A*, *Figure 7—figure supplement 1*). The invasive carcinoma samples from the remaining four patients exhibited a luminal phenotype, with elevated expression of luminal markers (*Figure 7A*, *Figure 7—figure supplement 1*). Consistent with the data from our BBN model and human bladder cancer cell lines (*Figures 4* and *6*), the luminal phenotype of the human primary bladder cancer samples was accompanied by a marked increase in the expression of *SHH*, whereas human primary tumors with the basal phenotype showed a significant decrease in *SHH* expression (*Figure 7B*).

To further determine whether our findings in the BBN mouse model showing that the molecular subtype of bladder cancer is determined by the expression of *Shh* regulated by DNA methylation are valid in human urothelial carcinoma, we performed methylation analysis by bisulfite sequencing in six basal subtypes and three luminal subtypes of primary invasive carcinoma samples (*Table 1*), and then compared their methylation status to that of three benign urothelia. We found that the methylation of the *SHH* gene, especially in the CpG shore of the promoter region, was significantly increased in the basal subtype of human invasive carcinomas compared to that in benign urothelia

**Table 1.** Patient sources for the subtype analysis of invasive urothelial carcinoma samples.
Muscle-invasive bladder carcinoma samples were obtained from radical cystectomy or transurethral resection patients with available disease and treatment histories, as shown.

| # | Sex | Age | Tumor stage and grade | Tissue source | Intravesical therapy | Neoadjuvant chemotherapy | Recurrence |
|---|-----|-----|----------------------|---------------|---------------------|--------------------------|------------|
| 1 | M | 65 | T4a (High) N0 | TURB | N | N/A | Y |
| 2 | M | 61 | T4a (High) N2 | Cystectomy | N | N | N |
| 3 | M | 56 | T2 (High) | TURB | N | N/A | N |
| 4 | M | 61 | T2 (High) | TURB | N | N/A | N |
| 5 | F | 74 | T2 (High) | TURB | N | N/A | Y |
| 6 | M | 59 | T1 (High) | TURB | BCG | N/A | Y |
| 7 | M | 74 | T1 (High) | TURB | N | N/A | N |
| 8 | M | 62 | T1 (High) N0 | TURB | N | N/A | N |
| 9 | M | 59 | T3 (High) | TURB | N | N/A | Y |
| 10 | F | 49 | T2 (High) N0 | TURB | N | N/A | N |

DOI: https://doi.org/10.7554/eLife.43024.021

(*Figure 7C,D*). Decreased methylation in luminal subtypes of invasive carcinoma relative to basal subtypes was also observed in these human samples (*Figure 7C,D*). Our analysis of primary human bladder cancer samples for the expression and associated methylation status of *SHH* was consistent with our findings from the murine BBN model of bladder cancer.

As the basal subtype of human urothelial carcinoma is the most aggressive form of bladder cancer, associated with lower median overall survival (*Choi et al., 2014*), we sought to determine whether the *SHH* expression level was associated with clinical outcome in human bladder cancer patients. We analyzed patient outcomes among a set of 41 muscle-invasive bladder tumors from Seoul National University Hospital (*Table 2*) that were clustered into two groups, showing low *SHH* expression (n = 31) or high *SHH* expression (n = 10). We found that the group with high *SHH* expression survived for a median of 50 months, compared to 39 months in the group with low *SHH* expression. The patient group with higher *SHH* expression therefore had better patient outcomes, presenting a 28% survival benefit compared to the group with lower *SHH* expression (log rank test p<0.05, *Figure 7E*).

In addition, we analyzed data from recently published large-scale studies of gene expression in human invasive carcinoma (*Robertson et al., 2017*) to assess the association of *SHH* expression with the molecular subtypes and patient survival. First, we analyzed the clinical outcomes of two subgroups stratified by Hh and BMP pathway genes (*Figure 6—figure supplement 1D*). We noted that the group with low *SHH* expression showed molecular characteristics of the basal subtype with poor patient outcomes compared to those of the high-*SHH* group with luminal characteristics (*Figure 6—figure supplement 1E,F*). Based on unsupervised hierarchical clustering, we further identified three distinct subgroups (*Figure 7F*): basal, luminal, and p53-like subtypes, as previously reported (*Choi et al., 2014*; *Robertson et al., 2017*). We noted that the basal cluster showed the lowest level of *SHH* expression (*Figure 7F*), consistent with our previous analysis and results from other studies (*Robertson et al., 2017*; *Shin et al., 2014b*), and the median overall survival of patients with the basal subtype was significantly lower than that of patients with the luminal subtype (log rank test p<0.0001, *Figure 7G*). It is important to note that only a small fraction of the patients carried genetic mutations in *SHH* and its downstream genes, suggesting that genetic alterations might not have been the major cause of the loss of *SHH* expression signatures in human invasive bladder cancer (*Figure 1—figure supplement 1A*). These results support our findings from the BBN mouse model and the analysis of primary human samples showing that the expression of *Shh* was regulated by epigenetic modifications, rather than mutational changes during the development and growth of invasive urothelial carcinoma (*Figures 1*, *6* and *7*).

Taken together, our findings from extensive analyses of primary tumor samples and a TCGA dataset suggest a possible basis for the development of the basal subtype of human invasive carcinomas

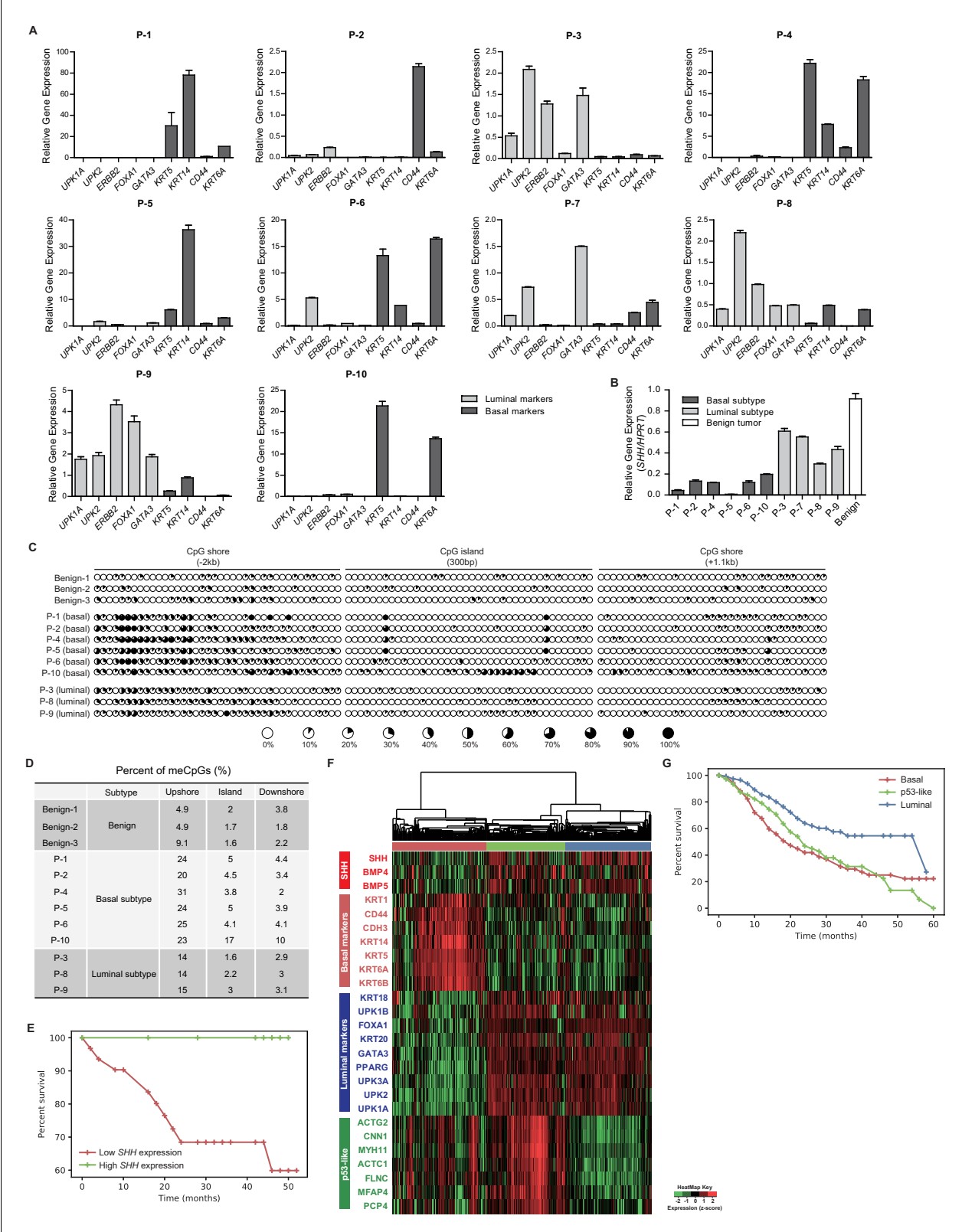

**Figure 7.** Expression analysis of patient-derived urothelial carcinomas and large-scale transcriptional analyses. (**A**) The relative expression of basal (*KRT5*, *KRT14*, *CD44* and *KRT6A*) and luminal markers (*UPK1A*, *UPK2*, *ERBB2*, *FOXA1* and *GATA3*) was analyzed in human invasive urothelial carcinomas from 10 patients. Data are presented as the mean ± SEM. n = 3 technical replicates (**B**) Expression of *SHH* in benign urothelium (white bar) and two subtypes of invasive urothelial carcinomas (basal, dark grey bars; luminal, light grey bars) from patients. Data are presented as the mean ± SEM. n = 3
*Figure 7 continued on next page*

*Figure 7 continued*

technical replicates. (C) The methylation status of the CpG island and CpG shore regions of the human *SHH* gene was analyzed by bisulfite sequencing in human invasive urothelial carcinoma tissues from patients (three benign tissues, six basal tumors, and three luminal tumors). The average degree of methylation is indicated by the black portion of the white circle. (D) The results obtained from bisulfite sequencing analysis of (C) are summarized. (E) Kaplan-Meier survival curve from a set of 41 patients with muscle-invasive bladder cancer from Seoul National University Hospital. Patients were classified by the expression level of *SHH* (high in green, n = 10; low in red, n = 31). The high- and low-*SHH* groups were determined by an *SHH*/18S value greater or less than 0.288, respectively. (F) Stratification of gene expression in RNA-seq data from the TCGA database of muscle-invasive urothelial carcinoma. Heatmap showing the expression levels (z-score normalized log2 (FPKM +1) values) of *SHH*, basal markers, luminal markers and p53-like markers. Based on unsupervised hierarchical clustering, three subgroups were identified: basal (red cluster, n = 125), p53-like (green, n = 106) and luminal (blue, n = 118). (G) Survival analysis of patients in the three subgroups. Basal (n = 125), p53-like (n = 106) and luminal (n = 118) subtypes are indicated by red, green and blue, respectively. The basal subtype was associated with lower median survival (27.79 months) than the luminal subtype (40.04 months) and p53-like subtype (28.73 months). Note that the basal subtype exhibited a significantly shorter life-span than the luminal subtype (log rank p<0.0001). See also *Figure 6—figure supplement 1D,E,F,G* and *Tables 1* and *2*.

DOI: https://doi.org/10.7554/eLife.43024.022

The following source data and figure supplement are available for figure 7:

**Source data 1.** Expression analysis of patient-derived urothelial carcinomas and large-scale transcriptional analyses.

DOI: https://doi.org/10.7554/eLife.43024.024

**Figure supplement 1.** Histopathological analysis of patient-derived urothelial carcinomas.

DOI: https://doi.org/10.7554/eLife.43024.023

that lack active Hh signaling. Under this scenario, loss of *SHH* expression due to hypermethylation leads to the decreased expression of stromal BMP, which in turn stimulates the formation of the basal subtype of human bladder cancer, with poor clinical outcomes.

## Discussion

The proposed events during the development of urothelial carcinoma, progressing from Shh-positive basal stem cells to distinct molecular subtypes, are summarized in *Figure 8*. Following the formation of widespread CIS, the clonal loss of Shh expression caused by DNA methylation leads to suppression of the stromal Hh response, which in turn increases the number of undifferentiated premalignant cells with proliferative advantages due to the decreased expression of stromal differentiation factors. This critical event facilitates the initiation and progression of tumors at early stages, ultimately resulting in the formation of the Shh-negative, basal subtype of invasive carcinoma. This model is strongly supported by our observations of halted initiation at the early stage and significantly decreased progression to invasive carcinoma following an increased stromal response to Hh signal upon the treatment with a potent DNMT inhibitor to epigenetically increase the expression of *Shh*. Once formed, the basal subtype of urothelial carcinoma remains Shh negative, consistent with the observed loss of *Shh* in murine and human invasive urothelial carcinoma with basal characteristics (*Cancer Genome Atlas Research Network, 2014a*; *Cancer Genome Atlas Research Network, 2014b*; *Shin et al., 2014b*). In contrast, invasive urothelial carcinoma of the luminal subtype develops through the distinct papillary lineage, as Shh expression is maintained, with a low CIS mutational profile. One key feature of our model is that a luminal-like subtype can be induced from the basal subtype of urothelial carcinoma by pharmacological modulation with 5'-azacitidine, to increase the expression of Shh in tumor cells, resulting in an increased stromal Hh response. This observation raises the possibility of additional pathways leading to the development of the luminal subtype of bladder cancer; a gain of Shh expression due to epigenetic plasticity in the basal subtype induces the luminal subtype of urothelial carcinoma. The Hh-dependent conversion of tumor subtypes is likely mediated by stromal Bmps, as supported by our observation that increased *Bmp* expression in tumor cells dramatically facilitates basal-to-luminal transition.

Epigenetic modifications, such as DNA methylation (*Herman and Baylin, 2003*) and histone modifications (*Seligson et al., 2005*) of genes associated with carcinogenesis, are commonly detected in human malignancies (*Feinberg and Tycko, 2004*; *Jones and Baylin, 2002*). Our previous findings of the invariable loss of Shh in murine and human urothelial carcinomas (*Shin et al., 2014b*), together with studies of other solid tumors, strongly support a role of Shh as a tumor suppressor during cancer progression (*Lee et al., 2014*; *Rhim et al., 2014*; *Yang et al., 2017*), raising the important questions concerning the molecular mechanisms by which the expression of Shh is lost in tumor cells at

**Table 2.** Patient information for survival analysis.

| # | Sex | Age | Tumor stage and grade | Tissue source | Intravesical therapy | Neoadjuvant chemotheraphy | Recurrence | Death |
|---|-----|-----|----------------------|---------------|---------------------|---------------------------|------------|-------|
| 1 | M | 54 | T1 | TURB | BCG | N/A | Y | Y |
| 2 | M | 60 | Ta (Low) | TURB | MMC | N/A | Y | N |
| 3 | M | 55 | Ta (High) | TURB | BCG | N/A | Y | N |
| 4 | M | 51 | Ta (Low) | TURB | BCG | N/A | Y | N |
| 5 | M | 76 | Ta (High) | TURB | BCG | N/A | N | N |
| 6 | M | 71 | Ta (High) | TURB | MMC | N/A | N | N |
| 7 | M | 58 | Ta (Low) | TURB | x | N/A | N | N |
| 8 | M | 83 | Ta (High) | TURB | BCG | N/A | N | N |
| 9 | M | 71 | T3a (High) N2 | TURB | BCG | N | Y | Y |
| 10 | M | 76 | T1 (High) | TURB | N | N/A | N | N |
| 11 | M | 80 | T1 (High) | TURB | N | N/A | Y | Y |
| 12 | M | 73 | T1 (High) | TURB | BCG | N/A | Y | N |
| 13 | M | 85 | T1 (High) | TURB | BCG | N/A | Y | N |
| 14 | M | 84 | T1 (High) | TURB | BCG | N/A | N | N |
| 15 | M | 83 | T1 (High) | TURB | BCG | N/A | N | N |
| 16 | M | 79 | T1 (High) | TURB | BCG | N/A | N | N |
| 17 | M | 65 | T4a (High) N0 | TURB | N | N | Y | Y |
| 18 | M | 84 | T1 (High) | TURB | N | N/A | N | Y |
| 19 | M | 79 | Tis N0 | TURB | N | N | N | N |
| 20 | M | 68 | T1 N0 | TURB | N | Y | N | N |
| 21 | M | 69 | T3b (High) N0 | TURB | N | N | N | Y |
| 22 | M | 86 | T2 (High) | TURB | N | N/A | Y | N |
| 23 | M | 80 | T3b (High) N0 | TURB | N | N | N | Y |
| 24 | M | 70 | T3b (High) N0 | TURB | N | Y | N | Y |
| 25 | M | 61 | T0 N0 | Cystectomy | BCG | Y | Y | N |
| 26 | M | 65 | T1 (High) N0 | Cystectomy | N | N | N | N |
| 27 | M | 53 | T2b (High) N0 | TURB | N | N | N | N |
| 28 | M | 62 | T3b (High) N2 | Cystectomy | N | N | N | Y |
| 29 | M | 70 | T1 N1 | TURB | N | N | N | N |
| 30 | M | 43 | T2a (High) N2 | Cystectomy | Gemcitabine/Cisplatin, BCG | N | Y | N |
| 31 | M | 60 | T2a (High) N0 | TURB | N | N | N | N |
| 32 | F | 66 | T1 (High) N2 | Cystectomy | N | N | N | N |
| 33 | M | 66 | Tis N0 | TURB | N | Y | N | N |
| 34 | M | 59 | T3 (High) | Cystectomy | N | N | Y | N |
| 35 | F | 57 | T0 N0 | TURB | N | N | N | N |
| 36 | F | 49 | T2b (High) N0 | TURB | N | N | N | N |
| 37 | M | 77 | Tis N0 | TURB | N | N | N | N |
| 38 | M | 72 | T0 N0 | TURB | N | N | N | Y |
| 39 | F | 76 | T2a (High) N0 | Cystectomy | N | N | N | N |
| 40 | F | 75 | T3a (High) N0 | TURB | N | N | N | N |
| 41 | F | 77 | T3b (High) N0 | Cystectomy | N | N | N | N |

DOI: https://doi.org/10.7554/eLife.43024.025

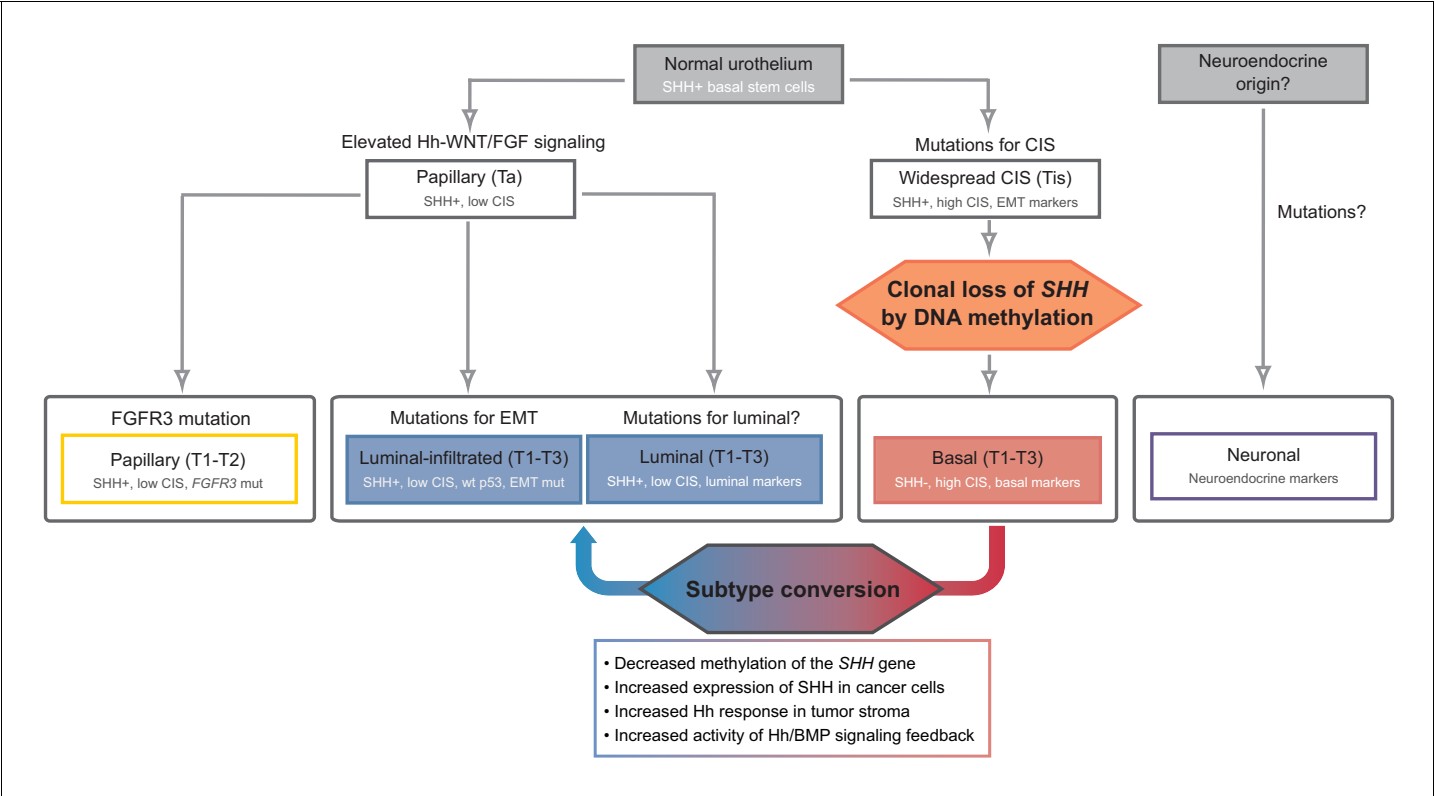

**Figure 8.** Model of the development of human urothelial carcinoma, progressing from SHH-positive basal stem cells to distinct molecular subtypes. Clonal loss of *SHH* expression caused by DNA methylation leads to the suppression of the stromal Hh response, which in turn facilitates the initiation and progression of tumors at early stages, ultimately resulting in the formation of the SHH-negative, basal subtype of invasive carcinoma. Decreased methylation of the *SHH* gene suppresses tumor growth by inducing subtype conversion from basal to luminal-like urothelial carcinoma through increased activity of Hh/BMP signaling feedback between cancer cells and the tumor stroma. See text.
DOI: https://doi.org/10.7554/eLife.43024.026

early stages of carcinoma development. A recent TCGA analysis of human bladder cancers revealed frequent mutations in genes associated with epigenetic regulation, but interestingly, showed no mutational changes in the *SHH* gene, suggesting the possibility that epigenetic changes are responsible for the loss of Shh expression during the development of urothelial carcinomas. Our study reveals that hypermethylation of the CpG shore, the upstream region with a reduced CG density of the CpG island in the *Shh* promoter region, is responsible for the loss of Shh, and that pharmacological inhibition of DNA methylation with agents such as 5'-azacitidine induces increased expression of Shh, thereby halting the progression of urothelial carcinoma. These findings are supported by other studies that show the importance of the CpG shore regions, rather than CpG islands, of genes involved in tumorigenesis (*Ellis et al., 2009*; *Irizarry et al., 2009*). It is, however, still not clear regarding the potential upstream regulators that control *Shh* methylation during the development of urothelial carcinoma. A previous TCGA analysis showed frequent mutations in methyltransferases and demethylases, such as *MLL2*, *ARID1A*, and *KDM6A*, raising the interesting possibility that the increased *Shh* methylation is due to the mutations in these genes associated with epigenetic regulation. Further studies will be necessary to investigate the specific roles of these genetic changes in the epigenetic regulation of the *Shh* gene. Furthermore, it would be interesting to determine whether other epigenetic events regulating chromatin organization, such as histone modifications, are also involved in the loss of Shh and whether the complex enhancer of the *Shh* gene, a cluster of three long-range, 840 kb regions (*Lettice et al., 2003*; *Sagai et al., 2009*), is associated with the development of urothelial carcinoma.

One of the most interesting findings of the present study is the remarkable ability of tumors to change their phenotypic characteristics without significant alterations in their genetic profiles.

Conventionally, cancer has been considered as a disease of 'sequential accumulation of mutations', and tumor phenotypes are dictated by their genetic alterations. However, recent advances in cancer biology and genomics have modified this traditional view, demonstrating that cancer is not a single-factorial genetic disease and that tumors exhibit a great degree of plasticity. Our observation that the tumor subtypes are epigenetically switched through the dynamic interactions with the tumor stroma—mediated by Hh-BMP signaling feedback—provides a clear basis for this recent view regarding tumor plasticity. In the normal bladder, our previous study has indicated that the development and regeneration of the urothelium depend on the balanced proliferation of basal stem cells and their differentiation into luminal cells, which is controlled by the stromal production of multiple factors (*Shin et al., 2011*). It is interesting to speculate that in cases of bladder tumors, the hierarchical relationship between basal stem cells and luminal cells observed in the normal bladder is maintained as basal subtype is converted into luminal subtype of bladder cancer. Interestingly, all luminal-like tumors induced by 5'-azacitidine treatment in our study do not show histological features of the basal subtype (e.g. squamous differentiation), although some luminal-like tumors still maintain the low expression of certain basal markers. This suggests the interesting possibility that there may be the intermediate subtype between basal and luminal human bladder cancer, similar to the intermediate cells in the urothelium of the normal bladder. Moreover, in our rescue experiment, we noticed that suppressing Hh-BMP feedback signaling with individual shRNAs for *Shh* or *Bmpr1a* along with 5'-azacitidine treatment was insufficient to regain the gene expression of the strong basal signature based on the GSEA analyses (*Figure 5D,E* and *Figure 5—figure supplement 1F,G*), although the reverted tumors show increased expression of certain basal markers (e.g. CK5) and the strong histological characteristic of the basal subtype (e.g. squamous differentiation) (*Figure 5B*). These data imply the potential involvement of additional mechanisms regulated by DNA methylation—other than Hh-BMP feedback signaling—to support the complete conversion of bladder tumors into the luminal subtype, particularly the complete loss of the basal signature, because the tissue dynamics of normal urothelium are controlled by multiple stromal factors such as Wnt, Fgf, and Bmp (*Shin et al., 2011*; *Shin et al., 2014b*). Further studies are required to investigate this possibility. Taken together, our findings on subtype conversion provided further insights on the current understanding of tumor plasticity and tumor interaction with other tissue components for the development of more effective therapeutic interventions for urothelial carcinomas.

Early detection and prevention are important therapeutic options for the treatment of cancers (*Tomasetti et al., 2017*; *Tomasetti and Vogelstein, 2015*), and a better understanding of tumor initiation and progression at early stages is crucial for the development of the most effective prevention strategies for cancer therapy. Pharmacological manipulation with 5'-azacitidine to increase Shh expression, resulting in therapeutic activation of stromal Bmp to block bladder tumor initiation through the differentiating of premalignant cells, could have a potentially beneficial impact on the clinical management of human bladder cancer, especially at the early stage of onset or recurrence, which makes this malignancy difficult and more costly to treat than other cancers. In addition to clinical benefits in terms of cancer prevention, our findings regarding the tumor-restraining effects of the stromal Hh response induced with pharmacological agents in full-fledged tumors, carry significant therapeutic implications, especially for patients with late-stage of invasive urothelial carcinoma. A recent TCGA analysis revealed five different subtypes of muscle-invasive urothelial carcinoma with different patient outcomes (*Cancer Genome Atlas Research Network, 2014a*; *Choi et al., 2014*; *Hedegaard et al., 2016*; *Robertson et al., 2017*). In these works, human invasive urothelial carcinomas were subdivided into five distinct molecular subtypes: luminal-papillary, luminal-infiltrated, luminal, basal and neuronal. Our results show that the loss of Shh induced by DNA methylation of the CpG shore of the *Shh* promoter region leads to the initiation and, ultimately, growth of the basal subtype of urothelial carcinoma. With pharmacological induction of the expression of Shh in epithelial tumor cells of the basal subtype, which in turn increases the stromal Hh response, urothelial carcinoma develops into a less aggressive, more manageable luminal-like subtype of bladder cancer through stromally produced Bmp-mediated cellular differentiation. The luminal subtype of urothelial carcinoma appears to grow more slowly, to be associated with better patient outcomes and to be more susceptible to the immune system than the basal subtype (*Figures 2* and *3*). Interestingly, a recent study showed a phenotypic shift of luminal parental tumors to basal subtypes in patient-derived organoid culture, and showed that this change is reversed in organoid xenografts in which the tumor stroma is restored (*Lee et al., 2018*). In the present study, we observed epigenetically

regulated Hh expression in cancer cells, which in turn induces a stromal response in the tumor micro-environment to determine the molecular subtype of bladder cancer. Our findings of Hh-dependent subtype switching through the tumor stroma may provide a strong rationale for the phenotypic plasticity observed in the human bladder tumor organoid model, which lacks a tumor stroma. The ability to convert molecular subtypes by modulation of Hh signaling with pharmacological agents such as 5'-azacitidine could therefore facilitate the development of personalized therapeutic options for urothelial carcinoma and permit potential combinatorial therapy. For example, treatment with 5'-azacitidine and/or FK506 (*Shin et al., 2014b*; *Spiekerkoetter et al., 2013*) to activate the Hh response-induced expression of cellular differentiation factors, such as BMPs, in the tumor stroma would induce the luminal-like subtype, which is more manageable with subsequent treatment, including immunotherapy (*Massard et al., 2016*; *Powles et al., 2014*).

In the present study, we speculate that several additional key events may be associated with Hh signaling during the initiation and growth of urothelial carcinomas not belonging to the basal and luminal subtypes, for which the mechanisms of progression are described above (*Figure 8*). Non-muscle-invasive carcinoma, for example, is derived from Shh-positive basal epithelial cells, which have been previously identified as cells of origin of bladder cancer (*Shin et al., 2014a*), via a papillary pathway by maintaining active Hh signaling, in which Shh is expressed in the epithelium and the Hh response occurs in the stroma. The Hh response in the stroma induces the expression of secreted factors such as Wnts and Fgfs, which in turn stimulate the proliferation of epithelial cells (*Shin et al., 2011*). With proliferative advantage of increased signaling feedback between the epithelium and stroma and the additional mutations of genes such as *Fgfr3*, papillary invasive tumors develop while maintaining Shh expression and low CIS signatures. The luminal–infiltrated subtype also follows the papillary lineage, but rather than mutations in *Fgfr3*, it accumulates mutations associated with epithelial-mesenchymal transition (EMT) while maintaining a moderate level of Shh expression and showing no mutations in *p53*. Finally, the neuronal subtype characterized by the expression of neuroendocrine and neuronal genes (*Sjödahl et al., 2017*) may originate from neuroendocrine cells, but not basal stem cells, and may develop in an Hh signaling-independent way. This speculation arises from the results of our bioinformatic analysis of the TCGA data for the levels of *SHH* and *BMP* in five distinct subtypes of invasive urothelial carcinoma (*Figure 6—figure supplement 1G*). Taken together, our study identifying several key events and their association with Hh signaling during the development of different subtypes of bladder cancer provides the basis for the design of precise therapeutic strategies for individual patients with genetic variability.

# Materials and methods

**Key resources table**

| Reagent type (species) or resource | Designation | Source or reference | Identifiers | Additional information |
|---|---|---|---|---|
| Strain, strain background (*M. musculus*) | C57BL/6J | The Jackson Laboratory | JAX:000664, RRID:IMSR_JAX:000664 | |
| Genetic reagent (*M. musculus*) | Col1a2$^{CreER}$ | The Jackson Laboratory | JAX:029235, RRID:IMSR_JAX:029235 | |
| Genetic reagent (*M. musculus*) | Smo$^{flox/flox}$ | The Jackson Laboratory | JAX:004526, RRID:IMSR_JAX:004526 | |
| Genetic reagent (*M. musculus*) | Gli2$^{flox/flox}$ | The Jackson Laboratory | JAX:007926, RRID:IMSR_JAX:007926 | |
| Genetic reagent (*M. musculus*) | Bmpr1a$^{flox/flox}$ | *Mishina et al., 2002* | MMRRC:030469-UNC, RRID:MMRRC_030469-UNC | |

*Continued on next page*

*Continued*

| Reagent type (species) or resource | Designation | Source or reference | Identifiers | Additional information |
|---|---|---|---|---|
| Genetic reagent (*M. musculus*) | CAnN.Cg-Foxn1nu/Crl | Charles River | CRL:194, RRID:IMSR_CRL:194 | |
| Genetic reagent (*M. musculus*) | NSG (NOD-*scid* IL2Rgamma$^{null}$) | The Jackson Laboratory | JAX:005557, RRID:IMSR_JAX:005557 | |
| Cell line (*H. sapiens*) | J82 | ATCC | ATCC: HTB-1, RRID:CVCL_0359 | |
| Cell line (*H. sapiens*) | T24 | ATCC | ATCC: HTB-4, RRID:CVCL_0554 | |
| Cell line (*H. sapiens*) | TCC-SUP | ATCC | ATCC: HTB-5, RRID:CVCL_1738 | |
| Antibody | Anti-Cytokeratin 5 (rabbit polyclonal) | Abcam | Abcam: ab53121, RRID:AB_869889 | IHC (1:300) |
| Antibody | Anti-Cytokeratin 8/18 (mouse monoclonal) | Developmental Studies Hybridoma Bank | DSHB: TROMA-I, RRID:AB_531826 | IHC (1:300) |
| Antibody | Ki67 antibody - Proliferation Marker | Abcam | Abcam ab15580, RRID:AB_443209 | IHC (1:500) |
| Recombinant DNA reagent | pCMV.dR 8.74 | | | Packaging plasmid |
| Recombinant DNA reagent | pMD2.G | Addgene | RRID:Addgene_12259 | Envelope plasmid |
| Recombinant DNA reagent | pSicoR-mCh-empty | Addgene | RRID:Addgene_21907 | PMID:19587682 |
| Recombinant DNA reagent | pSiCoR | Addgene | RRID:Addgene_11579 | PMID:15240889 |
| Recombinant DNA reagent | pSicoR-mCh-mShh | This paper | | Lentiviral vector expressing shRNA targeting the murine *Shh* gene |
| Recombinant DNA reagent | pSicoR-mCh-hSHH | This paper | | Lentiviral vector expressing shRNA targeting the human *SHH* gene |
| Recombinant DNA reagent | pSicoR-mCh-mBmpr1a | This paper | | Lentiviral vector expressing shRNA targeting the murine *Bmpr1a* gene |
| Recombinant DNA reagent | pSicoR-mCh-hBMPR1A | This paper | | Lentiviral vector expressing shRNA targeting the murine *BMPR1A* gene |
| Recombinant DNA reagent | Puro.Cre empty vector | Addgene | RRID:Addgene_17408 | PMID:18308936 |

*Continued on next page*

*Continued*

| Reagent type (species) or resource | Designation | Source or reference | Identifiers | Additional information |
|---|---|---|---|---|
| Recombinant DNA reagent | pLenti6.3/V5-Bmp4 | This paper | | Lentiviral vector expressing murine Bmp4 |
| Peptide, recombinant protein | Recombinant Murine BMP-4 | PeproTech | PeproTech: 315–27 | |
| Commercial assay or kit | MethylEdge Bisulfite Conversion System | Promega | Promega: N1301 | |
| Commercial assay or kit | TaKaRa EpiTaq HS (for bisulfite-treated DNA) | TaKaRa | TaKaRa: R110A | |
| Commercial assay or kit | pGEM-T Easy Vector System I | Promega | Promega: A1360 | |
| Commercial assay or kit | RNeasy Mini Kit | Qiagen | Qiagen: 74104 | |
| Commercial assay or kit | DNeasy Blood and Tissue Kit | Qiagen | Qiagen: 69504 | |
| Commercial assay or kit | QIAshredder | Qiagen | Qiagen: 79654 | |
| Commercial assay or kit | High-Capacity cDNA reverse transcription kit | Applied biosystem | Applied biosystem: 4368814 | |
| Commercial assay or kit | Power SYBR Green PCR Master Mix | Applied biosystem | Applied biosystem: 4367706 | |
| Chemical compound, drug | N-Butyl-N-(4-hydroxybutyl) nitrosamine | Tokyo Chemical Industry | TCI: B0938 | |
| Chemical compound, drug | 5-aza-2'-deoxycytidine | Sigma | Sigma: A3656 | |
| Chemical compound, drug | Mirus Bio TransIT-LT1 Transfection Reagent | Mirus Bio | Mirus Bio: MIR 2300 | |
| Chemical compound, drug | Corning Matrigel Growth Factor Reduced (GFR) | Corning Life Science | Corning: 354230 | |
| Chemical compound, drug | Corning Matrigel Basement Membrane Matrix High Concentration (HC) | Corning Life Science | Corning: 354248 | |
| Chemical compound, drug | polybrene (hexadimethrine bromide) | Sigma | Sigma: H9286 | |
| Chemical compound, drug | Blasticidin | Gibco | Gibco: R21001 | |
| Chemical compound, drug | Puromycin dihydrochloride | Sigma | Sigma: P8833 | |
| Software, algorithm | Image J | ImageJ | RRID:SCR_003070 | (http://imagej.nih.gov/ij/) |

*Continued on next page*

*Continued*

| Reagent type (species) or resource | Designation | Source or reference | Identifiers | Additional information |
|---|---|---|---|---|
| Software, algorithm | GraphPad Prism | GraphPad Prism | RRID:SCR_015807 | Version 6, https://graphpad.com |
| Software, algorithm | SnapGene Viewer | Snap Gene | RRID:SCR_015053 | http://www.snapgene.com/products/snapgene_viewer/ |
| Software, algorithm | MUSCLE | EMBL-EBI | RRID:SCR_011812 | http://www.ebi.ac.uk/Tools/msa/muscle/ |
| Software, algorithm | Methprimer 2.0 | The Li Lab at UCSF, *Li and Dahiya, 2002* | RRID:SCR_010269 | http://urogene.org/ |
| Software, algorithm | Oasis2 | *Han et al., 2016* | | https://sbi.postech.ac.kr/oasis2/ |
| Software, algorithm | Java Treeview | *Saldanha, 2004* | RRID:SCR_016916 | http://jtreeview.sourceforge.net/ |
| Software, algorithm | Cluster 3.0 | *de Hoon et al., 2004* | | http://bonsai.hgc.jp/~mdehoon/software/cluster/software.htm |
| Software, algorithm | Oncoprint | *Gao et al., 2013* ; *Cerami et al., 2012* | RRID:SCR_014555 | http://www.cbioportal.org/ |
| Software, algorithm | Tophat | *Trapnell et al., 2009* | | |
| Software, algorithm | Cufflinks | *Trapnell et al., 2012* | | |
| Software, algorithm | GSEA | Broad Institute | RRID:SCR_003199 | http://www.broadinstitute.org/gsea/index.jsp |

## Mice

For the genetic deletion experiments, *Col1a2*$^{CreER}$ (RRID:IMSR_JAX:029235) mice were mated with the *Smo*$^{flox/flox}$ (RRID:IMSR_JAX:004526) or *Gli2*$^{flox/flox}$ (RRID:IMSR_JAX:007926) strains to obtain *Col1a2*$^{CreER}$;*Smo*$^{flox/flox}$, *Col1a2*$^{CreER}$;*Gli2*$^{flox/flox}$ mice, which were administered 8 mg of TM (Sigma) per 30 g of body weight on 3 consecutive days by oral gavage. Male mice at 8–10 weeks of age were used. For experiment involving 5'-azacitidine, mice were injected intraperitoneally with 1 mg of 5'-azacitidine (Sigma) per 1 kg of body weight daily. The duration of the dosing is described in each figure. In each experiment, mice in each cage were randomly selected for drug/TM or control treatments. Mouse procedures were performed under isoflurane anesthesia with a standard vaporizer. All procedures were performed under a protocol approved by the Institutional Animal Care and Use Committee at POSTECH (IACUC number: POSTECH-2017–0094).

## BBN-induced bladder carcinogenesis

A 0.1% concentration of BBN (TCI) was dissolved in drinking water, and BBN-containing water was provided to mice *ad libitum* for 4 to 6 months in a dark bottle. BBN-containing water was changed twice a week. Bladders were collected and analyzed after 4 to 6 months of BBN administration.

## Methylation analysis using bisulfite sequencing of genomic DNA

The DNA methylation status of murine and human *Shh* was determined using bisulfite genomic DNA sequencing. For bisulfite conversion, 1 µg of genomic DNA was converted using the MethylEdge Bisulfite Conversion System (Promega), following the manufacturer's instructions. The genomic sequence of the regulatory region of murine and human *Shh* was obtained from the NCBI nucleotide database (Mus musculus: NC_000071.6, Homo sapiens: NG_007504.2), and the CpG island and CpG shores in the regulatory region were identified by Methprimer 2.0 (*Li and Dahiya, 2002*) (RRID:SCR_010269). The 2 kb regions upstream and downstream of the CpG island were referred to as the 'CpG upshore' and 'CpG downshore' regions, respectively. For sequencing analysis, bisulfite-converted DNA was amplified by EpiTaq HS (TaKaRa), and PCR products containing the CpG island and CpG shore regions were subcloned into the pGEM-T easy vector (Promega). The region containing the CpG island and shores was divided into eight sub-regions, and each sub-region was amplified using specific primers designed for bisulfite-converted target sequences (summarized in *Table 3*). The sequencing data were assembled using SnapGene software (https://snapgene.com/, RRID:SCR_015053) and the MUSCLE: multiple sequence alignment tool (https://www.ebi.ac.uk/Tools/msa/muscle/, RRID:SCR_011812). The average degree of methylation was obtained from the analysis of 8–10 clones of each sub-region. The methylated CpG sites were counted and distinguished from the unmethylated CpG sites.

## Bladder organoid culture

BBN-induced bladder tumors were minced and then incubated in DMEM (Gibco) containing collagenase I, II (20 mg/ml each) and thermolysin (250 KU/ml) at 37°C for 2 hr, with 5 min trituration every 30 min. A single-cell suspension was obtained and filtered through 100 µm cell strainers (Falcon). After lysis of red blood cells in ACK lysis buffer (Gibco), the cells were washed with DMEM containing 10% fetal bovine serum (Millipore) and counted using a hemocytometer (Sigma). For bladder organoid culture, single tumor cells were overlaid in growth factor-reduced Matrigel (Corning) and incubated with advanced DMEM/F-12 (Gibco) supplemented with 10 mM HEPES (pH 7.4, Sigma), 10 mM Nicotinamide (Sigma), 1 mM N-acetyl-L-cysteine (Sigma), GlutaMAX (Gibco), 1% penicillin/streptomycin (Gibco), 50 ng/mL mouse EGF (Peprotech), 0.5X B-27 (Gibco), 1 µM A8301, and 10 µM Y-27632. For Bmp4 treatment, organoids were treated with recombinant Bmp4 protein (Peprotech) for 8 days, with medium transition every 2 days. For the knock-down experiments, bladder tumor

**Table 3.** Information for primers used in the bisulfite sequencing of regulatory regions of the murine and human *SHH* genes.

| Target species | Primer name | Forward sequence (5'-3') | Reverse sequence (5'-3) |
|---|---|---|---|
| Mouse | *Shh* promoter | TTTTTAGTTTTGTTATTATTTAAAATTAGG | CAAAAATCACCAAAAAACATCTAAC |
| | *Shh* upshore region 1 | TTTGTATATTTATATTTGGGGATGG | AAAAAACTTATAAAACAAACTACCTTTC |
| | *Shh* upshore region 2 | TTGTATTTTGTTAGGATAGATTGGAAG | ACCCCATCCCCAAATATAAATATAC |
| | *Shh* upshore region 3 | GGATGGTGAGGTTTTGTTATATTGT | ATATCCAACACTCTTTCAAAAAAAA |
| | *Shh* upshore region 4 | TTGAAGTAAAATGAGGTTTTAGGATGT | CACCATCCCAAACTTAAAAAAATTA |
| | *Shh* downshore region 1 | ATGTTGTTGTTGTTGGTTAGATGTT | ATAAAAAACCCCATCTTCTAATACC |
| | *Shh* downshore region 2 | GGGTATTAGAAGATGGGGTTTTTTA | CCCAAACTTTCTCAATTACAATTCT |
| | *Shh* downshore region 3 | GAAAGTTTGGGGGTAGTTTTGATA | TATTTACAAAAAAACCCATTTCCAA |
| Human | *SHH* promoter | TTTTTTTGTTTTTTGATTGTTGTTT | TCAACTTTTTAAAATACCTCCTCTTC |
| | *SHH* upshore region 1 | TTTTGGGGAAGAAAAATTAAATAAT | CAACAATCAAAAAACAAAAAAAATCTA |
| | *SHH* upshore region 2 | AGTGAGGTGATTATAGATTTAAAGAT | CAACTATTATTTAATTTTTCTTCCCC |
| | *SHH* upshore region 3 | ATTTGTAAAGGGAATTTTTGGAAAT | AACCAAAAAAATAAAATTTAAAACTCC |
| | *SHH* upshore region 4 | TGTTAAGGGTGGAAGGTAGGGTAGTT | CAAAAATTCCCTTTACAAATCAACT |
| | *SHH* downshore region 1 | GGAAGAGGAGGTATTTTAAAAAGTTG | AACTAAACCCTTAACCTCCATTCTC |
| | *SHH* downshore region 2 | GAGAATGGAGGTTAAGGGTTTAGTT | CCTCCTAACTTTTCCAATTAAAAAT |
| | *SHH* downshore region 3 | ATTTTTAATTGGAAAAGTTAGGAGG | CAAAAAAAACCCATTTCTAACTTCAA |

DOI: https://doi.org/10.7554/eLife.43024.027

organoids were infected with lentivirus containing shRNAs specific for mouse/human *Shh* or *Bmpr1a* (*Table 4*). For lentivirus production, 293 T cells were co-transfected with lentiviral plasmids and packaging vectors (pCMV.dR 8.74 and pMD2.G [RRID:Addgene_12259], Addgene) using TransIT-LT1 (Mirus Bio). The supernatant was collected 48 hr post-transfection and filtered through 0.45-μm-pore PES filter (Millipore). The viral titer was calculated in 3T3 cells by serial dilution of the virus-containing supernatant. For lentiviral infection, bladder organoids were incubated in lentivirus-containing medium with 8 μg/ml polybrene (Sigma) for 12 hr at 37°C. For selection of infected organoids, GFP- or mCherry-positive organoids were picked up from Matrigel under a fluorescence microscope.

## Orthotopic transplantation of bladder tumors

Bladder tumors were dissociated into single cells as described above. Cells were resuspended in 80 μl DMEM containing 50% Matrigel (BD Bioscience) and then submucosally injected with 29-gauge insulin syringes into the anterior aspect of the bladder dome. The abdominal incisions and skin were then closed with a 4–0 nylon suture, and the surgical site was disinfected with alcohol. For orthotopic transplantation of tumor organoids, bladder tumor organoids were selected, resuspended in 50% organoid medium and 50% Matrigel and then transplanted into recipient mice.

## Human bladder tumor samples and cancer cell lines

Frozen human bladder tissue samples were obtained from the tissue bank of Seoul National University Hospital. For fresh bladder tumor samples, 0.5–1 cm$^3$ specimens of fresh bladder tissue were obtained from patients undergoing cystectomy or TURB under a protocol approved by the SNUH Institutional Review Board (IRB Number: 1607-135-777). Informed consent and consent to publish was obtained from the patients. The cancer tissues were evaluated before being transported to POSTECH for further analysis. For experiments involving bladder cancer cell lines, J82 (RRID:CVCL_0359), T24 (RRID:CVCL_0554) and TCC (RRID:CVCL_1738) were used. All cell lines were authenticated using STR profiling method and were tested negative for mycoplasma contamination.

## Quantitative RT-PCR

Human or mouse bladder samples were snap frozen in liquid nitrogen, then homogenized with a mortar and pestle, and RNA was extracted using the RNeasy Plus Mini Kit (Qiagen). The RNA samples were subsequently dissolved in RNase-free water, and their concentration and purity were determined with a spectrophotometer. The TAE/formamide electrophoresis method (*Masek et al., 2005*) was used for the analysis of RNA quality. For quantitative RT-PCR of mRNA transcripts, first-strand cDNA was synthesized using a High-Capacity cDNA reverse Transcriptase kit (Applied Biosystems) with oligo dT. Quantitative RT-PCR was performed using SYBR Green Supermix (Applied Biosystems) and a One-step cycler (Applied Biosystems), and gene expression was normalized to the housekeeping gene *HPRT1*.

## Histological analysis

Tumor specimens were prefixed in 10% neutral-buffered formalin for 12 hr, then embedded in paraffin and sectioned into 4-μm-thick sections using a microtome. The slides were stained with hematoxylin and then counter-stained with eosin for histological analysis. For immunostaining, tumor specimens were embedded in OCT compound (Tissue-Tek) and sectioned into 10-μm-thick sections with a cryostat (Leica).

**Table 4.** shRNA sequences targeting the murine and human *SHH* and *BMPR1A* genes.

| Target species | shRNA | Target sequence | Sense | Antisense |
|---|---|---|---|---|
| Mouse | *Bmpr1a* | GGGTCGTTACAACCGTGATTT | GGGUCGUUACAACCGUGAUUU | AAAUCACGGUUGUAACGACCC |
| | *Shh* | CTTTAGCCTACAAGCAGTTTA | CUUUAGCCUACAAGCAGUUUA | UAAACUGCUUGUAGGCUAAAG |
| Human | *BMPR1A* | GTCCAGATGATGCTATTAATA | GUCCAGAUGAUGCUAUUAAUA | UAUUAAUAGCAUCAUCUGGAC |
| | *SHH* | CTACGAGTCCAAGGCACATAT | CUACGAGUCCAAGGCACAUAU | AUAUGUGCCUUGGACUCGUAG |

DOI: https://doi.org/10.7554/eLife.43024.028

## Immunofluorescence analysis of tissue sections

Bladder tumors were dissected from mice, fixed in 10% neutral-buffered for 3 hr, washed three times in PBS, incubated in 30% sucrose overnight, and embedded in OCT compound (Tissue-Tek). The slides were subsequently washed twice in PBS, blocked in 2% goat serum with 3% BSA in PBS containing 0.25% Triton X-100 for 1 hr, and incubated overnight at 4°C in a humidified chamber with primary antibodies diluted in blocking solution. Sections were washed three times with PBS containing 0.25% Triton X-100 and incubated for 1 hr at RT with the appropriate Alexa Fluor-conjugated secondary antibodies diluted 1:1000 in blocking solution. The slides were next washed three times with PBS and tissue sections were mounted with Prolong Gold mounting reagent (Invitrogen). All immunofluorescence images were analyzed by confocal microscopy (Leica SP5 or Olympus FV1000).

## RNA-Seq library construction

Total RNA was extracted with TRIzol reagent (Thermo Fisher) according to the manufacturer's instructions. RNA-seq libraries were constructed using the TruSeq sample Prep Kit V2 (Illumina). Quantity of RNA-seq library was determined by Nanodrop, and average quantity of RNA-seq libraries ranged from 30 to 50 ng/µl. RNA-seq libraries were sequenced using a NextSeq platform with single-end reads of 75 bases.

## Differential gene expression and gene set enrichment analysis (GSEA) of RNA-seq data

Raw reads from fastq files were aligned to the reference mouse genome assembly mm10 using Tophat allowing no mismatch (*Trapnell et al., 2009*). Differentially expressed genes were analyzed using Cufflinks tools (*Trapnell et al., 2012*). From all annotated genes, genes were removed if the average of rpkm across all sequenced samples is below 1.0, likely to have low depth to assign the genes. Gene set enrichment analyses were performed according to the instructions (RRID:SCR_003199). To generate a custom gene set for each luminal and basal marker, representative genes for each signature were obtained from previous study (*Damrauer et al., 2014*). The RNA-seq datasets used in the study have been deposited in NCBI GEO (Accession number: GSE129441).

## Data analysis

Statistical analysis was performed using GraphPad Prism software v.6. (RRID:SCR_015807). All data are presented as the mean ± SEM, and two group comparisons were conducted with a two-tailed Student's test. A value of $p<0.05$ was considered statistically significant. For the analysis of TCGA data, the gene expression levels of muscle-invasive bladder cancer patients were downloaded from the TCGA data portal (https://portal.gdc.cancer.gov/). The FPKM expression values were $\log_2(x + 1)$ transformed for convenient comparison of mRNA abundance estimates, where x denotes the FPKM value for each gene. The log-transformed expression values were normalized to z-scores for further analysis. Gene Cluster 3.0 was used for unsupervised hierarchical clustering (*de Hoon et al., 2004*), with default settings of an uncentered correlation and centroid linkage for the similarity metric and clustering method, respectively. Visualization of the mRNA cluster results was conducted using Java TreeView (*Saldanha, 2004*) (RRID:SCR_016916). To examine the clinical output of different mRNA clusters, survival analysis was conducted using the Oasis2 tool (*Han et al., 2016*). For the Kaplan-Meier survival plot, patients with overall survival of 5 years or less were considered for survival analysis. The Oncoprint format of mutation occurrences was plotted from cBioPortal (*Cerami et al., 2012*; *Gao et al., 2013*) (RRID:SCR_014555).

## Acknowledgements

We thank Phil Beachy for helpful discussion regarding this manuscript. This research was supported by grants from the National Research Foundation of Korea to KS: NRF-2017R1A2B4006043, NRF-2017M3C7A1047875, NRF-2017R1A5A1015366 and the BK21 Plus Program.

# Additional information

## Funding

| Funder | Grant reference number | Author |
|---|---|---|
| National Research Foundation of Korea | NRF-2017R1A2B4006043 | Kunyoo Shin |
| National Research Foundation of Korea | NRF-2017M3C7A1047875 | Kunyoo Shin |
| National Research Foundation of Korea | NRF-2017R1A5A1015366 | Kunyoo Shin |
| National Research Foundation of Korea | Brain Korea 21 Program for Leading Universities & Students | Kunyoo Shin |

The funders had no role in study design, data collection and interpretation, or the decision to submit the work for publication.

## Author contributions

SungEun Kim, Conceptualization, Resources, Data curation, Formal analysis, Validation, Investigation, Visualization, Methodology, Writing—original draft, Project administration, Writing—review and editing; Yubin Kim, Resources, Data curation, Validation, Investigation, Visualization, Methodology, Writing—original draft, Writing—review and editing; JungHo Kong, Resources, Data curation, Software, Formal analysis, Investigation, Visualization, Writing—original draft, Project administration, Writing—review and editing; Eunjee Kim, Jae Hyeok Choi, Formal analysis, Validation, Investigation, Methodology; Hyeong Dong Yuk, Resources, Data curation, Formal analysis, Investigation, Methodology; HyeSun Lee, Resources, Data curation, Investigation, Methodology; Hwa-Ryeon Kim, Resources, Data curation, Software, Formal analysis, Investigation, Visualization; Kyoung-Hwa Lee, Resources, Supervision, Investigation, Methodology; Minyong Kang, Data curation, Formal analysis, Investigation, Project administration, Writing—review and editing; Jae-Seok Roe, Resources, Data curation, Software, Formal analysis, Supervision, Investigation, Visualization, Project administration, Writing—review and editing; Kyung Chul Moon, Resources, Formal analysis, Investigation, Methodology; Sanguk Kim, Conceptualization, Resources, Software, Supervision, Writing—original draft, Project administration, Writing—review and editing; Ja Hyeon Ku, Conceptualization, Resources, Supervision, Investigation, Writing—original draft, Project administration, Writing—review and editing; Kunyoo Shin, Conceptualization, Resources, Data curation, Software, Formal analysis, Supervision, Funding acquisition, Visualization, Methodology, Writing—original draft, Project administration, Writing—review and editing

## Author ORCIDs

Sanguk Kim (ID) https://orcid.org/0000-0002-3449-3814
Kunyoo Shin (ID) https://orcid.org/0000-0002-1519-9839

## Ethics

Human subjects: Frozen human bladder tissue samples were obtained from the tissue bank of Seoul National University Hospital. For fresh bladder tumor samples, 0.5-1 cm$^3$ specimens of fresh bladder tissue were obtained from patients undergoing cystectomy or TURB under a protocol approved by the SNUH Institutional Review Board (IRB Number: 1607-135-777). Informed consent and consent to publish was obtained from the patients.
Animal experimentation: Mouse procedures were performed under isoflurane anesthesia with a standard vaporizer. All procedures were performed under a protocol approved by the Institutional Animal Care and Use Committee (IACUC) at POSTECH (POSTECH-2017-0094).

## Decision letter and Author response

Decision letter https://doi.org/10.7554/eLife.43024.036

Author response https://doi.org/10.7554/eLife.43024.037

---

## Additional files

### Supplementary files

• Transparent reporting form
DOI: https://doi.org/10.7554/eLife.43024.029

### Data availability

All data generated or analysed during this study are included in the manuscript and supporting files. Source data files have been provided for Figures 1, 3, 4, 5, 6 and 7.

The following dataset was generated:

| Author(s) | Year | Dataset title | Dataset URL | Database and Identifier |
|---|---|---|---|---|
| Roe J, Kunyoo Shin | 2019 | Epigenetic regulation of mammalian Hedgehog signaling to the stroma determines the molecular subtype of human bladder cancer | https://www.ncbi.nlm.nih.gov/geo/query/acc.cgi?acc=GSE129441 | NCBI Gene Expression Omnibus, GSE129441 |

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
