## [Decision Letter]

Thank you for submitting your article "Epigenetic regulation of Hedgehog signaling to the stroma determines the molecular subtype of bladder cancer" for consideration by *eLife*. Your article has been reviewed by three peer reviewers, one of whom is a member of our Board of Reviewing Editors, and the evaluation has been overseen by Jeffrey Settleman as the Senior Editor. The following individual involved in review of your submission has agreed to reveal their identity: David McConkey (Reviewer #3).

The reviewers have discussed the reviews with one another and the Reviewing Editor has drafted this decision to help you prepare a revised submission.

Summary:

This manuscript examines the role of SHH signaling in mouse models of chemically-induced bladder carcinogenesis, and shows that inhibition of DNA methylation results in decreased tumorigenesis. The authors report that this effect is mediated by a HH response in the stroma, and that inhibition of BMP signaling can promote basal subtype differentiation of the tumors.

These findings are potentially important and are of particular interest to bladder cancer biologists as well as the broader community of cancer biologists. In particular, the new findings indicating a central role for epithelial-stromal interactions in the regulation of bladder tumor growth and subtype specification are novel and significant. However, there are substantial gaps in the experimental analysis as well as methodology that need to be addressed before this manuscript might be suitable for publication in *eLife*. In addition, the manuscript would benefit from a more thoughtful discussion of the findings. However, overall, the manuscript is of generally high quality presenting carefully designed and well executed experiments that have been carefully interpreted.

Essential revisions:

1) The nature of the epigenetic changes and the cells in which they occur are poorly characterized. It is assumed that the SHH methylation occurs in the epithelium and that 5'-azacytidine acts in the epithelium to promote the expression of SHH in the BBN bladder tumors. However, this has not been specifically demonstrated. Since the tumor organoids lack stroma, the changes observed in the organoids are presumed to be similar to those in the tumors, but the levels of SHH in the organoids appear to be much lower, raising concerns as to whether the pathways are analogous. Similarly, the stromal cells that respond to SHH signaling and express BMP4 are not identified, and the expression of BMP4 in tumor stroma is not examined. Hence, what is the nature of the stromal cells that respond to HH signaling? Can the authors exclude the possibility that 5'-azacytidine has off-target effects in the stroma that are responsible for the observed phenotypes?

2) Another major issue is the lack of detailed analysis of basal and luminal subtypes, which the authors identify on the basis of qPCR analysis of a small number of markers. These subtypes should be established by analysis of RNA-Seq expression profiles and comparison with standard signatures (e.g., Damrauer et al., 2014). Such an analysis would be particularly helpful in evaluating the status of control tumors that express both CK5 and CK18, which the authors describe as luminal markers, but may in fact have an intermediate phenotype. It would also be useful to evaluate the luminal/basal subtype status of in vivo tumors such as those in Figure 1, in addition to the tumor grafts.

3) There are additional issues that arise with the analysis of basal and luminal subtypes. In particular, the authors should address whether the basal subtype tumors display squamous differentiation and the extent to which these tumors evaluated by a trained pathologist. Higher-magnification images of these tumors would be useful to show whether the tumors have a squamous differentiation phenotype. Most importantly, can the authors demonstrate that the luminal-to-basal subtype conversion is due to phenotypic changes in the tumor cells, and not due to competition between clones of different phenotypes within the tumors?

4) The manuscript would be greatly enhanced by discussion of major implications of the findings reported in the current study. What are the potential epigenetic regulators that control SHH methylation during bladder tumorigenesis? What are the implications of the HH-dependent tumor subtype conversion for bladder cancer progression? At present, the authors' Discussion skirts these key issues, and largely reiterates the findings described in the Results such that the Discussion section is largely a restatement of the Results section.

[Editors' note: further revisions were requested prior to acceptance, as described below.]

Thank you for resubmitting your work entitled "Epigenetic regulation of mammalian Hedgehog signaling to the stroma determines the molecular subtype of bladder cancer" for further consideration at *eLife*. Your revised article has been favorably evaluated by Jeffrey Settleman as the Senior Editor, a Reviewing Editor, and two reviewers.

The manuscript has been improved but there are some remaining issues that need to be addressed before acceptance, as outlined below:

It is essential that the authors please address the comments of reviewer #2 regarding:

1) The appropriate interpretation of the GSEA analysis as outlined below;

2) The authors should also consider revising their Discussion section to discuss the implications of subtype conversion for bladder cancer progression, especially since there is still controversy in the field as to whether and how frequently this occurs and;

3) Finally, the authors should describe the steps to be taken to make their RNA-Seq datasets available through an appropriate publicly accessible database.

*Reviewer #2:*

There are remaining issues regarding the response to the query about analysis of RNA-seq expression profiles and comparison with standard signatures (Essential revision #2). The authors have performed RNA-seq analyses as requested, but the Gene Set Enrichment Analysis using luminal and basal signatures is confusingly described. While it is definitely the case that GSEA with the small signatures used is often difficult to interpret, the data shown in Figure 5D and E clearly indicate that there is no positive enrichment with the basal signature. Consequently, these data do not seem consistent with the newly added statements in the text that "Consistent with these results, our analysis of RNA-seq expression profiles revealed the strong basal signature with relatively low expression of lunimal markers in the tumors expressing shRNA for Shh" and "Consistent with these results, our analysis of RNA-seq expression profiles revealed the strong basal signature in the tumors expressing shRNA for Bmpr1a". In these cases, the staining patterns for Ck5 shown in Figure 5B may not accurately reflect the overall basal status of these tumors. In addition, the actual basal and luminal gene sets used for these analyses are unclear, since Damrauer et al., 2014 is cited in the subsection “Differential gene expression and gene set enrichment analysis (GSEA) of RNA-seq data”, but Dadhania et al., 2016 is cited in the subsection “Heightened activity of Hh signaling to the stroma induces a less aggressive luminal subtype of urothelial carcinoma”.

It is also disappointing that the Discussion has not been extensively revised to provide more insight into the reported findings (Essential revision #4). In particular, the authors still do not discuss the implications of subtype conversion for bladder cancer progression, especially since there is still controversy in the field as to whether and how frequently this occurs. The newly added finding in response to Essential revision #3 that most of the basal tumors display squamous differentiation is not discussed at all.

Finally, the authors do not mention whether their RNA-seq datasets have been deposited in an appropriate publicly accessible database.

---

## [Author Response]

Essential revisions:1) The nature of the epigenetic changes and the cells in which they occur are poorly characterized. It is assumed that the SHH methylation occurs in the epithelium and that 5'-azacytidine acts in the epithelium to promote the expression of SHH in the BBN bladder tumors. However, this has not been specifically demonstrated. Since the tumor organoids lack stroma, the changes observed in the organoids are presumed to be similar to those in the tumors, but the levels of SHH in the organoids appear to be much lower, raising concerns as to whether the pathways are analogous. Similarly, the stromal cells that respond to SHH signaling and express BMP4 are not identified, and the expression of BMP4 in tumor stroma is not examined. Hence, what is the nature of the stromal cells that respond to HH signaling? Can the authors exclude the possibility that 5'-azacytidine has off-target effects in the stroma that are responsible for the observed phenotypes?

In response to the reviewers’ concerns regarding the physiological relevance of our tumor organoids, we have conducted the experiments in more optimized conditions that precisely mimic the *in vivo* tumor environment. In these revised experiments, we treated tumor organoids with 5’-azacytidine (5’-Aza), 7 days after initial seeding of tumor cells. The results showed that Shh expression levels were similar to those in the *N*-butyl-*N*-(4-hydroxybutyl)-nitrosamine (BBN) tumor (Figure 1F, data newly added), indicating that our tumor organoid model accurately recapitulates *in vivo* tumor growth.

Regarding the nature of cells that respond to the Shh ligand to express Bmp4, our previous study clearly demonstrated that Shh secreted by the urothelium activates the Hh response in stromal fibroblasts, resulting in stromal expression of Bmp4 (Shin et al., 2014; Figure 3, 4). In this study, we employed stromal-specific ablation of the Hh response using Gli1CreER;Smo^flox/flox^ and observed decreased expression of Bmp4 when the stromal Hh response was attenuated (Shin et al., 2014; Figure 3B). In addition to mice, Bmp4 levels increase only in stromal fibroblasts upon treatment with the Hh pathway agonists or ShhN protein in humans (Shin et al., 2014; Figure 4B). Furthermore, in this study, similar to that in our previous study, we utilized mouse strains that can specifically ablate the Hh response in stromal fibroblasts for transplantation experiments of Bmp4-expressing tumor organoids. The results showed that the growth of these tumor organoids significantly reduced (Figure 3A, B), thereby demonstrating a reciprocal signal feedback of the Hh and Bmp pathways between the epithelium and the stroma.

To further address the question regarding direct demonstration of a reciprocal epithelial–stromal signal feedback loop, we performed additional experiments in which we co-cultured mouse and human tumor organoids with cancer-associated fibroblasts (Author response image 1). Upon treatment of tumor organoids with 5’-Aza, we found that the Hh pathway activation occurred only in stromal fibroblasts, as indicated by an increase in Gli1 expression (Author response image 1). Concomitant with the pathway activation, we found that Bmp4 levels increased.

**Author response image 1. respfig1:** Co-culture experiments of tumor organoids with cancer-associated fibroblasts. (**A**) Schematic diagram for the experiments designed to demonstrate epithelial-stromal Hh/Bmp feedback loop. (**B**) Expression of Gli1, Bmp4, and Shh in mouse/human tumor organoids or cancer-associated fibroblasts treated with 5-Aza as described in (**A**). (**C**) Schematic diagram for the experiments to test off-target effects of 5-Aza in the stroma. (**D**) Expression of Shh in mouse/human tumor organoids cultured with or without cancer-associated fibroblasts as described in (**C**).

Considering the reviewers’ question of specific action of 5’-Aza on tumor cells, we treated tumor organoids with 5’-Aza in the presence or absence of cancer-associated fibroblasts (Author response image 1) to determine whether the drug has off-target effects in the stroma. The results demonstrated that even in the absence of cancer-associated fibroblasts, Shh expression in tumor organoids increases to a similar level as that observed in control experiments with cancer-associated fibroblasts (Author response image 1). These finding indicated that at the dose utilized, 5’-Aza has no off-target effects on the stroma to indirectly affect Shh expression in tumors. Taken together with our previous study, these new data provide strong evidence of cancer-associated fibroblasts as stromal cells that respond to Shh to express Bmp4. Furthermore, our additional experimental results indicate that Shh methylation occurs in tumor cells and that Aza acts specifically through epithelial tumor cells.

2) Another major issue is the lack of detailed analysis of basal and luminal subtypes, which the authors identify on the basis of qPCR analysis of a small number of markers. These subtypes should be established by analysis of RNA-Seq expression profiles and comparison with standard signatures (e.g., Damrauer et al., 2014). Such an analysis would be particularly helpful in evaluating the status of control tumors that express both CK5 and CK18, which the authors describe as luminal markers, but may in fact have an intermediate phenotype. It would also be useful to evaluate the luminal/basal subtype status of in vivo tumors such as those in Figure 1, in addition to the tumor grafts.

We conducted RNA sequencing (RNA-seq) expression profile analysis, as suggested by the reviewer. The new data show RNA-seq analysis of bladder tumors from animals which were orthotopically transplanted with BBN-induced tumors in the presence or absence of 5’-Aza treatment (Figure 4G, Figure 4—figure supplement 1G, data newly added) and rescue animals which were orthotopically transplanted with tumor organoids containing shRNA for Shh or Bmpr1 (Figure 5D, E, Figure 5—figure supplement 1F, G, data newly added). The results indicated that all basal tumors show a clear standard signature of the basal subtype. For the luminal subtype, consistent with our immunostaining data, we found strong expression of luminal markers with relatively low expression of a few basal markers. Based on the fact that luminal subtype still expresses a few basal markers, such as Ck5, we believe that using the phrase “luminal subtype” was too strong; therefore, in the revised manuscript, we have moderated it to “luminal-like” or “acquiring a luminal phenotype.”

For the BBN-induced endogenous tumor in Figure 1, we found that all tumors exhibit a basal signature, as previously reported (Fantini et al., 2018; Shin et al., 2014; Author response image 2).

**Author response image 2. respfig2:** Heat map for the expression of basal and luminal marker genes on BBN-induced bladder tumors.

3) There are additional issues that arise with the analysis of basal and luminal subtypes. In particular, the authors should address whether the basal subtype tumors display squamous differentiation and the extent to which these tumors evaluated by a trained pathologist. Higher-magnification images of these tumors would be useful to show whether the tumors have a squamous differentiation phenotype. Most importantly, can the authors demonstrate that the luminal-to-basal subtype conversion is due to phenotypic changes in the tumor cells, and not due to competition between clones of different phenotypes within the tumors?

We have extensively analyzed the histopathology of murine and human basal tumors associated with subtype conversion, and we received assistance from Prof. KyungChul Moon, an expert urologic pathologist, and Prof. Ja hyeon Ku, a urologist, for the analysis. The new analyses comprised histopathological examination of the basal subtype of bladder tumors from BBN-treated animals (Figure 1), animals orthotopically transplanted with BBN-induced tumors (Figure 2, 4), and rescue animals orthotopically transplanted with tumor organoids containing shRNA for Shh or Bmpr1 (Figure 5), as well as human bladder tumors (Figure 7—figure supplement 1, data newly added). The results showed that all of murine basal subtype tumors show high levels of squamous differentiation. We have provided higher-magnification images in the revised manuscript, as requested. For human samples, four of six samples with the basal subtype showed squamous differentiation, whereas all four luminal subtype tumors showed no signs of squamous differentiation (Figure 7—figure supplement 1, data newly added).

As we may have failed to point our clearly, we focused on basal-to-luminal conversion. As previously reported (Fantini et al., 2018; Shin et al., 2014), on the basis of the expression level of basal markers and the mutational profile, the invasive carcinomas produced in our BBN model are most similar to the basal subtype of human urothelial carcinoma (Figure 4—figure supplement 1C), which is the most aggressive form of bladder cancer (Choi et al., 2014). In this study, we suggest that this aggressive form of urothelial carcinoma can be changed to a less aggressive luminal-like subtype by augmentation of the Hh pathway activity.

With regard to the possibility of luminal tumors arising from luminal clones that might already exist in basal tumors, we did not observe any luminal clones in the BBN-induced tumors with the basal subtype (Author response image 3). In this experiment, BBN-induced tumors were dissociated into single cells, which were then cultured as tumor organoids. The resulting tumor organoids were derived from single clones originated from BBN-induced tumors. We extensively analyzed 30 clones of tumor organoids; we found no evidence of luminal characteristics in any tumor organoids, indicating that the development of the luminal subtype of urothelial carcinoma is due to the subtype conversion of the basal tumor and not due to the clonal expansion of existing luminal clones.

**Author response image 3. respfig3:** Clonal analysis of BBN-induced bladder tumor. (**A**) Schematic diagram of the experiment. (**B**) Representative images of single cell-derived tumor organoids cultured for 2, 4, 6, 8, and 16 days. (**C**) Clonal analysis of single cell-derived tumor organoids for luminal and basal subtype.

4) The manuscript would be greatly enhanced by discussion of major implications of the findings reported in the current study. What are the potential epigenetic regulators that control SHH methylation during bladder tumorigenesis? What are the implications of the HH-dependent tumor subtype conversion for bladder cancer progression? At present, the authors' Discussion skirts these key issues, and largely reiterates the findings described in the Results such that the Discussion section is largely a restatement of the Results section.

We have included several major implications of our findings in the Discussion section, such as (i) subtype-specific management of human bladder cancer at an early stage, (ii) clinical/therapeutic implications of Hh-dependent subtype conversion (fourth paragraph), and (iii) the proposed role of Hh-dependent subtype conversion for the progression of urothelial carcinoma with implications for the development of five distinct subtypes of human bladder cancer (last paragraph). In the revised manuscript, as suggested, we have provided more discussion on potential epigenetic regulators that might control Shh expression.

[Editors' note: further revisions were requested prior to acceptance, as described below.]

Reviewer #2:

There are remaining issues regarding the response to the query about analysis of RNA-seq expression profiles and comparison with standard signatures (Essential revision #2). The authors have performed RNA-seq analyses as requested, but the Gene Set Enrichment Analysis using luminal and basal signatures is confusingly described. While it is definitely the case that GSEA with the small signatures used is often difficult to interpret, the data shown in Figure 5D and E clearly indicate that there is no positive enrichment with the basal signature. Consequently, these data do not seem consistent with the newly added statements in the text that "Consistent with these results, our analysis of RNA-seq expression profiles revealed the strong basal signature with relatively low expression of lunimal markers in the tumors expressing shRNA for Shh" and "Consistent with these results, our analysis of RNA-seq expression profiles revealed the strong basal signature in the tumors expressing shRNA for Bmpr1a". In these cases, the staining patterns for Ck5 shown in Figure 5B may not accurately reflect the overall basal status of these tumors. In addition, the actual basal and luminal gene sets used for these analyses are unclear, since Damrauer et al., 2014 is cited in the subsection “Differential gene expression and gene set enrichment analysis (GSEA) of RNA-seq data”, but Dadhania et al., 2016 is cited in the subsection “Heightened activity of Hh signaling to the stroma induces a less aggressive luminal subtype of urothelial carcinoma”.

We agree with reviewer #2 that the Gene Set Enrichment Analysis in Figure 5 is unclear. To clarify this, we have shifted the GSEA panels with the basal signature from Figure 5D, E to Figure 5—figure supplement 1F, G. In addition, we have clarified the unclear statements in the text and have further discussed the interpretation of the GSEA analysis in the Discussion section.

In brief, the rescue experiments presented in Figure 5 clearly show the negative enrichment with the luminal signature in transplanted tumors containing shRNA for *Shh* or *Bmpr1* (new Figure 5D, E). This finding suggests that the Hh-Bmp signaling axis between the tumor and tumor stroma is required for the maintenance of gene expression of the luminal signature. However, as the reviewer indicated, we found that there is no positive enrichment with the basal signature in tumors containing shRNA for *Shh* or *Bmpr1* (new Figure 5—figure supplement 1F, G) although the tumors show increased expression of certain basal markers (e.g., Ck5) and the strong histological characteristic of the basal subtype (e.g., squamous differentiation) (Figure 5B). These findings demonstrated that the suppression of Hh-Bmp feedback signaling with individual shRNAs for *Shh* or *Bmpr1* along with 5’-azacitidine treatment was not sufficient to regain the gene expression of the strong basal signature, implying the potential involvement of additional mechanisms regulated by DNA methylation – other than Hh-Bmp feedback signaling – to support the complete conversion into the luminal subtype, particularly the complete loss of the basal signature. Further studies are required to investigate this interesting possibility. In addition, we have removed the citation of Dadhania et al., 2016 because the basal and luminal signatures used in the present study were adapted from the previous study performed by Damrauer et al., 2014.

It is also disappointing that the Discussion has not been extensively revised to provide more insight into the reported findings (Essential revision #4). In particular, the authors still do not discuss the implications of subtype conversion for bladder cancer progression, especially since there is still controversy in the field as to whether and how frequently this occurs. The newly added finding in response to Essential revision #3 that most of the basal tumors display squamous differentiation is not discussed at all.

As per the reviewer’s suggestion, we have further discussed the implications of subtype conversion for bladder cancer in the revised manuscript.

Finally, the authors do not mention whether their RNA-seq datasets have been deposited in an appropriate publicly accessible database.

The RNA-seq datasets used in the present study have been deposited in NCBI GEO (Accession number: GSE129441).